# Identification and characterisation of hypomethylated DNA loci controlling quantitative resistance in *Arabidopsis*

Leonardo Furci[1†], Ritushree Jain[1†], Joost Stassen[1], Oliver Berkowitz[2], James Whelan[2], David Roquis[3,4], Victoire Baillet[5], Vincent Colot[5], Frank Johannes[3,4], Jurriaan Ton[1]*

[1]P3 Centre for Plant and Soil Biology, Department of Animal and Plant Sciences, University of Sheffield, Sheffield, United Kingdom; [2]Department of Animal, Plant and Soil Science, ARC Centre of Excellence in Plant Energy Biology, La Trobe University, Melbourne, Australia; [3]Department of Plant Sciences, Technical University of Munich, Freising, Germany; [4]Institute for Advanced Study, Technical University of Munich, Garching, Germany; [5]Institut de Biologie de l'Ecole Normale Supérieure (IBENS), Ecole Normale Supérieure, Centre National de la Recherche Scientifique (CNRS), Institut National de la Santé et de la Recherche Médicale (INSERM), PSL Université Paris, Paris, France

**Abstract** Variation in DNA methylation enables plants to inherit traits independently of changes to DNA sequence. Here, we have screened an *Arabidopsis* population of epigenetic recombinant inbred lines (epiRILs) for resistance against *Hyaloperonospora arabidopsidis (Hpa)*. These lines share the same genetic background, but show variation in heritable patterns of DNA methylation. We identified four epigenetic quantitative trait loci (epiQTLs) that provide quantitative resistance without reducing plant growth or resistance to other (a)biotic stresses. Phenotypic characterisation and RNA-sequencing analysis revealed that *Hpa*-resistant epiRILs are primed to activate defence responses at the relatively early stages of infection. Collectively, our results show that hypomethylation at selected pericentromeric regions is sufficient to provide quantitative disease resistance, which is associated with genome-wide priming of defence-related genes. Based on comparisons of global gene expression and DNA methylation between the wild-type and resistant epiRILs, we discuss mechanisms by which the pericentromeric epiQTLs could regulate the defence-related transcriptome.

DOI: https://doi.org/10.7554/eLife.40655.001

*For correspondence:
j.ton@sheffield.ac.uk

†These authors contributed equally to this work

Competing interests: The authors declare that no competing interests exist.

## Introduction

Eukaryotic cytosine methylation plays an important role in the regulation of gene expression and genome stability. In plants, this form of DNA methylation occurs at three sequence contexts: CG, CHG and CHH, where H indicates any base except guanine (G) (*Vanyushin, 2006*; *Law and Jacobsen, 2010*). Patterns of plant DNA methylation in the plant genome can remain stable over multiple generations and influence heritable phenotypes (*Quadrana and Colot, 2016*). Recent evidence has suggested that reduced DNA methylation increases the responsiveness of the plant immune system (*Espinas et al., 2016*) This 'priming' of plant defence enables an augmented induction of defence-related genes after pathogen attack, causing increased levels of quantitative resistance (*Prime-A-Plant Group et al., 2006*; *Conrath et al., 2015*; *Martinez-Medina et al., 2016*; *Liégard et al., 2018*). In some cases, priming of defence-related genes is associated with post-translational histone modifications that mark a more open chromatin structure (*Jaskiewicz et al., 2011*; *Luna et al.,*

**eLife digest** In plants, animals and microbes genetic information is encoded by DNA, which are made up of sequences of building blocks, called nucleotide bases. These sequences can be separated into sections known as genes that each encode specific traits. It was previously thought that only changes to the sequence of bases in a DNA molecule could alter the traits passed on to future generations. However, it has recently become clear that some traits can also be inherited through modifications to the DNA that do not alter its sequence.

One such modification is to attach a tag, known as a methyl group, to a nucleotide base known as cytosine. These methyl tags can be added to, or removed from, DNA to create different patterns of methylation. Previous studies have shown that plants whose DNA is less methylated than normal ('hypo-methylated') are more resistant to plant diseases. However, the location and identity of the hypo-methylated DNA regions controlling this resistance remained unknown. To address this problem, Furci, Jain et al. studied how DNA methylation in a small weed known as *Arabidopsis thaliana* affects how well the plants can resist a disease known as downy mildew.

Furci, Jain et al. studied a population of over 100 *A. thaliana* lines that have the same DNA sequences but different patterns of DNA methylation. The experiments identified four DNA locations that were less methylated in lines with enhanced resistance to downy mildew. Importantly, this form of resistance did not appear to reduce how well the plants grew, or make them less able to resist other diseases or environmental stresses.

The results of further experiments suggested that reduced methylation at the four DNA regions prime the plant's immune system, enabling a faster and stronger activation of a multitude of defence genes across the genome after attack by downy mildew.

The next steps following on from this work are to investigate exactly how the four DNA regions with reduced methylation can prime so many different defence genes in the plant. Further research is also needed to determine whether it is possible to breed crop plants with lower levels of methylation at specific DNA locations to improve disease resistance, but without decreasing the amount and quality of food produced.

DOI: https://doi.org/10.7554/eLife.40655.002

*2012*). Additional evidence for epigenetic regulation of plant immunity has come from independent studies reporting that disease-exposed *Arabidopsis* produces progeny that expresses transgenerational acquired resistance (TAR), which is associated with priming of defence-related genes (*Luna et al., 2012*; *Slaughter et al., 2012*). Furthermore, *Arabidopsis* mutants that are impaired in the establishment or maintenance of DNA methylation mimic TAR-related priming without prior priming stimulus (*Lopez et al., 2011*; *Luna and Ton, 2012*; *López Sánchez et al., 2016*). By contrast, the hyper-methylated *ros1-4* mutant, which is impaired in active DNA de-methytation, is more susceptible to biotrophic pathogens, affected in defence gene responsiveness, and impaired in TAR (*López Sánchez et al., 2016*; *Yu et al., 2013*). Thus, DNA (de)methylation determines quantitative disease resistance by influencing the responsiveness of defence-related genes. However, causal evidence that selected hypomethylated DNA loci are responsible for the meiotic transmission of this form of quantitative disease resistance is lacking.

Epigenetic Recombinant Inbred Lines (epiRILs) have been developed with the aim to study the epigenetic basis of heritable plant traits (*Reinders et al., 2009*; *Johannes et al., 2009*). EpiRILs show little differences in DNA sequence, but vary substantially in DNA methylation. A commonly used population of epiRILs is derived from a cross between the Arabidopsis wild-type (Wt) accession Col-0 and the *decreased DNA methylation1-2* (*ddm1-2*) mutant (*Johannes et al., 2009*). The DDM1 protein is a chromatin re-modelling enzyme that provides DNA methyltransferase enzymes access to heterochromatic transposable elements (TEs) (*Jeddeloh et al., 1998*; *Brzeski and Jerzmanowski, 2003*; *Zemach et al., 2013*). Accordingly, the *ddm1-2* mutation causes loss of pericentromeric heterochromatin and reduced DNA methylation in all sequence contexts (*Kakutani et al., 1996*; *Ito et al., 2015*). Although the epiRILs from the *ddm1-2* x Col-0 cross do not carry the *ddm1-2* mutation, they contain stably inherited hypomethylated DNA regions from the *ddm1-2* parent, which are maintained up to 16 generations of self-pollination (*Johannes et al., 2009*; *Colomé-Tatché et al.,*

*2012*; *Latzel et al., 2013*). A core set of 123 epiRILs from this population at the eight generation of self-pollination in the wild-type (Wt) background has been characterized for differentially methylated region (DMR) markers, enabling linkage mapping of heritable hypomethylated loci controlling root growth, flowering and abiotic stress tolerance (*Liégard et al., 2018*; *Cortijo et al., 2014*; *Kooke et al., 2015*).

In this study, we have characterised the core set of 123 lines from the *ddm1-2* x Col-0 epiRIL population for resistance against the biotrophic downy mildew pathogen *Hyaloperonospora arabidopsidis* (*Hpa*) to search for heritable hypomethylated loci controlling disease resistance. We identified four of these epigenetic quantitative trait loci (epiQTLs), accounting for 60% of the variation in disease resistance. None of these epiQTLs were associated with growth impairment, indicating that the resistance does not incur major physiological costs on plant development. Further phenotypic characterisation and transcriptome analysis of selected *Hpa*-resistant epiRILs revealed that their resistance is associated with genome-wide priming of defence-related genes. Interestingly, bisulfite sequencing did not reveal defence regulatory genes inside the epiQTL regions that were simultaneously primed and hypomethylated, suggesting that DDM1-dependent DNA methylation at the epiQTLs *trans*-regulates the responsiveness of distant defence genes.

## Results

### Identification of epiQTLs controlling quantitative resistance against *Hpa*

To examine the role of DDM1-dependent DNA methylation in heritable disease resistance, 123 epiRILs from the *ddm1-2* x Col-0 cross were analysed for *Hpa* resistance and compared to siblings of the *ddm1-2* parent (*Figure 1a*, red), the Wt parent (Col-0), and five progenies thereof (*Figure 1a*, green). Leaves of 3-week-old plants were inoculated with *Hpa* conidiospores and then collected for trypan-blue staining at 6 days post-inoculation (dpi). Microscopic classification of leaves into four classes of *Hpa* colonisation (*Figure 1—figure supplement 1*) revealed 51 epiRILs with statistically enhanced levels of resistance compared to each susceptible Wt line (Pearson's Chi-squared tests, p<0.05). Of these, eight epiRILs showed similar levels of *Hpa* resistance as the *ddm1-2* line (*Figure 1a*, dark blue triangles; Pearson's Chi-squared test, p>0.05), whereas 43 epiRILs showed intermediate levels of resistance. To identify the epiQTL(s) responsible for the observed variation in *Hpa* resistance, the categorical classification of *Hpa* infection was converted into a single value numerical resistance index (RI; *Figure 1a*, bottom graph). Using a linkage map of stably inherited DMR markers (*Colomé-Tatché et al., 2012*) (*Supplementary file 1* dataset S1), interval mapping revealed four statistically significant epiQTLs on chromosomes I, II, IV and V (*Figure 1b*). The epiQTL on chromosome II had the highest logarithm of odds (LOD) value. For all epiQTLs, the DMR markers with the highest LOD scores ('peak markers') showed a positive correlation between *ddm1-2* haplotype and RI (*Figure 1c*), indicating that the hypomethylated haplotype from *ddm1-2* increases resistance against *Hpa*. A linear regression model to calculate the percentage of RI variance explained by each peak marker ($R^2(g)$) (*Cortijo et al., 2014*) confirmed that the DMR peak marker of the epiQTL on chromosome II had the strongest contribution to RI variation. Using an additive model, the combined contribution of all epiQTL peak markers to RI variation ($R^2(G)$) (*Cortijo et al., 2014*) was estimated at 60.0% (*Figure 1d*).

DNA methylation maintains genome stability by preventing transposition of TEs. In the Col-0 x *ddm1-2* epiRIL population, reduced methylation at the *ddm1-2* haplotype occurs predominantly at long transposons in heterochromatic pericentromeric regions (*Zemach et al., 2013*; *Colomé-Tatché et al., 2012*). Frequent transposition events in the epiRILs are nevertheless rare as most DNA hypomethylation occurs at relic transposons that have lost the ability to transpose, and the occurrence of independent transposition events at similar loci is extremely unlikely (*Colomé-Tatché et al., 2012*; *Matzke and Mosher, 2014*). However, it is possible that transposition events originating from the heavily hypomethylated *ddm1-2* parent were crossed into the population, resulting into shared transposition events (STEs) between multiple epiRILs, which could have contributed to variation in resistance. To account for this possibility, we compared the genomic DNA sequences of the four epiQTL intervals from 122 epiRILs (LOD drop-off = 2) for the presence of STEs in more than two epiRILs, using TE-tracker software (*Gilly et al., 2014*). This analysis revealed three STEs in the epiQTL

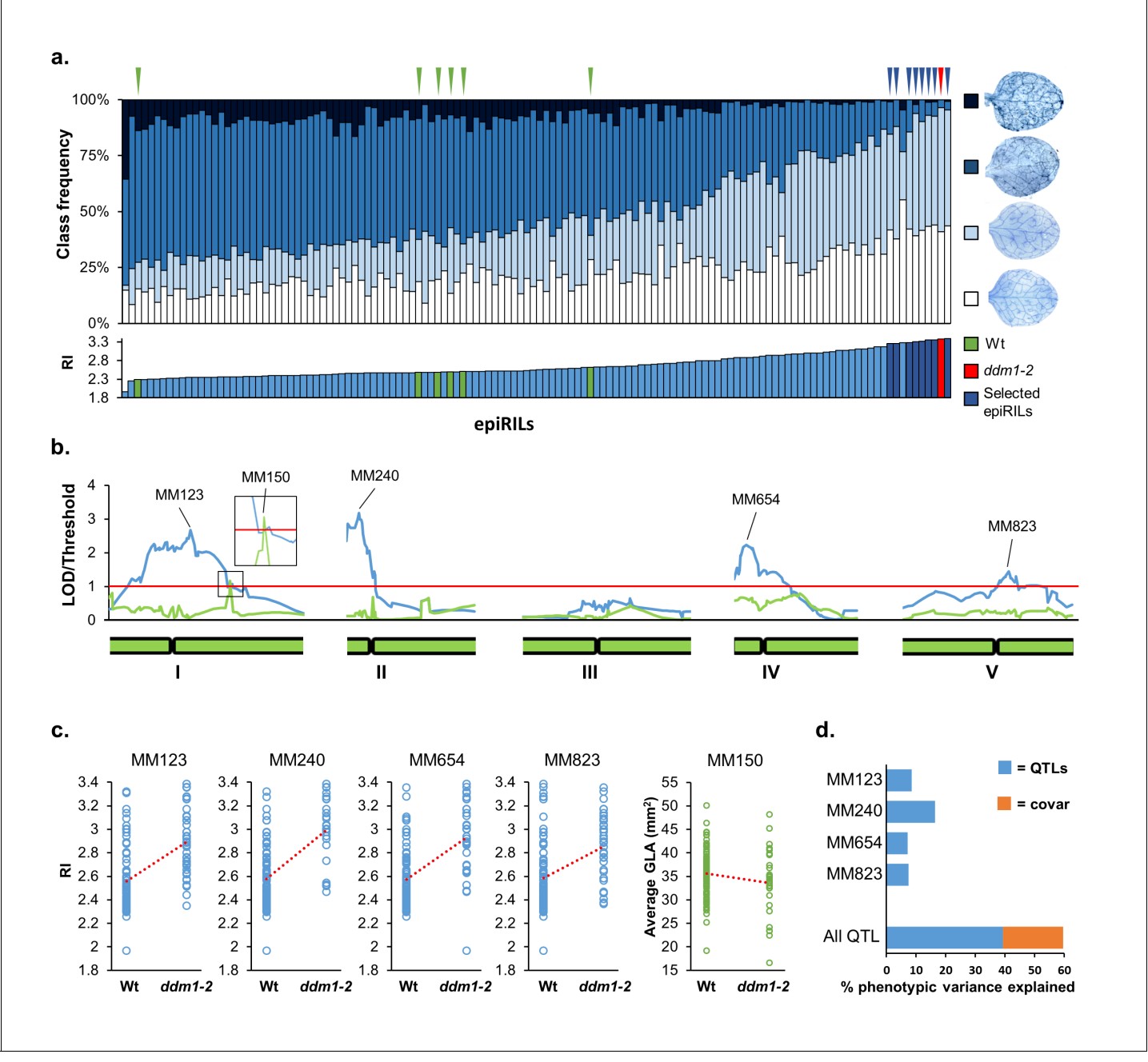

**Figure 1.** Mapping of epigenetic quantitative trait loci (epiQTL) controlling transgenerational resistance against *Hyaloperonosopra arabidopsidis* (*Hpa*). (a) Levels of *Hpa* resistance in 123 epiRIL lines, the *ddm1-2* line (F4; red triangle) and six Wt lines (Col-0; green triangles). Top graph shows distribution of infection classes in each epiRIL; blue triangles pinpoint the eight most resistant epiRILs with statistically similar levels of *Hpa* colonisation as the *ddm1-2* line (Pearson's Chi-squared test, p>0.05). Bottom graph shows variation in *Hpa* resistance index (RI). Green bars: Wt lines; red bar: *ddm1-2*; blue bars eight most resistant epiRILs (*n* > 100). (b) Linkage analysis of RI (blue line) and green leaf area (GLA) of three-week-old seedlings (green). Green bars at the bottom represent chromosomes. Red line represents the threshold of significance. Peak DMR markers with the highest LOD scores are shown on top. (c) Correlation plots between peak marker haplotype (methylated Wt versus hypomethylated *ddm1-2*) and RI (blue) or GLA (green). (d) Percentages of resistance variance explained by the peak DMR markers, including covariance between markers (orange).
DOI: https://doi.org/10.7554/eLife.40655.003

The following source data and figure supplements are available for figure 1:

**Source data 1.**
DOI: https://doi.org/10.7554/eLife.40655.013

**Figure supplement 1.** Representative examples of infection classes used for quantification of *Hpa* resistance.

*Figure 1 continued on next page*

*Figure 1 continued*

DOI: https://doi.org/10.7554/eLife.40655.004

**Figure supplement 2.** Average green leaf area (GLA) of the 123 epiRILs (light green), the *ddm1-2* line (F4; red) and six Wt lines (Col-0; dark green).

DOI: https://doi.org/10.7554/eLife.40655.005

**Figure supplement 2—source data 1.**

DOI: https://doi.org/10.7554/eLife.40655.006

**Figure supplement 3.** Resistance phenotypes of the eight most *Hpa*-resistant epiRILs against different (a) biotic stresses.

DOI: https://doi.org/10.7554/eLife.40655.007

**Figure supplement 3—source data 1.**

DOI: https://doi.org/10.7554/eLife.40655.008

**Figure supplement 4.** Defence marker phenotypes of the eight most *Hpa*-resistant lines.

DOI: https://doi.org/10.7554/eLife.40655.009

**Figure supplement 4—source data 1.**

DOI: https://doi.org/10.7554/eLife.40655.010

**Figure supplement 5.** Transgenerational stability of *Hpa* resistance in *Hpa*-resistant epiRILs.

DOI: https://doi.org/10.7554/eLife.40655.011

**Figure supplement 5—source data 1.**

DOI: https://doi.org/10.7554/eLife.40655.012

interval on chromosome I (*Supplementary file 1* dataset S2), while no STEs could be detected in the other epiQTL intervals. None of the STEs in the epiQTL on chromosome I showed statistically significant linkage with RI (*Supplementary file 1* dataset S2). Accordingly, we conclude that the segregating *Hpa* resistance in the epiRIL population is caused by epigenetic variation in DNA methylation, rather than genetic variation by STEs.

## Effects of the resistance epiQTLs on plant growth and resistance against other (a) biotic stresses

Expression of inducible defence mechanisms is often associated with physiological costs, resulting in reduced plant growth (*Huot et al., 2014*). To determine whether the resistance that is controlled by the four epiQTLs is associated with costs to plant growth, we quantified the green leaf area (GLA) of 12–15 individual plants per line at the stage of *Hpa* inoculation (*Figure 1—figure supplement 2*). Subsequent interval mapping revealed one statistically significant epiQTL on chromosome I (*Figure 1b*). The corresponding peak marker (MM150) showed a negative correlation between GLA and *ddm1-2* haplotype (*Figure 1c*), indicating that the hypomethylated *ddm1-2* allele at this locus represses plant growth. The growth epiQTL mapped to a different region than the resistance epiQTL on chromosome I (*Figure 1b*, inset). Furthermore, none of the eight most resistant epiRILs showed significant growth reduction compared to all Wt lines in the screen (*Figure 1—figure supplement 2*). Hence, the resistance provided by the four hypomethylated epiQTLs is not associated with major physiological costs to plant growth.

Enhanced defence to one stress can lead to enhanced susceptibility to another stress, which is caused by antagonistic cross-talk between defence signalling pathways (*Koornneef and Pieterse, 2008*). To examine whether *Hpa* resistance in the epiRIL population is associated with increased susceptibility to other stresses, we compared the eight most *Hpa*-resistant epiRILs (*Figure 1a*; *Figure 1—figure supplement 3a*) for resistance against the necrotrophic fungus *Plectosphaerella cucumerina* (*Pc*) and tolerance to salt (NaCl). At 9 dpi with *Pc* spores, epiRIL#193 showed a statistically significant reduction in necrotic lesion size compared to the Wt (line #602), indicating enhanced resistance (*Figure 1—figure supplement 3b*). The seven other epiRILs showed unaffected levels of *Pc* resistance that were similar to the Wt. Salt tolerance was quantified by the percentage of seedlings with fully developed cotyledons at 6 days after germination on agar medium with increasing NaCl concentrations. Remarkably, all *Hpa*-resistant epiRILs showed varying degrees of tolerance to the highest NaCl concentration compared to Wt plants (*Figure 1—figure supplement 3c*). Thus, the quantitative resistance to *Hpa* in the epiRIL population does not compromise resistance against necrotrophic pathogens or abiotic stress.

### *Hpa*-resistant epiRILs are primed to activate different defence mechanisms

Basal resistance against *Hpa* involves a combination of salicylic acid (SA)-dependent and SA-independent defence mechanisms (*Knoth et al., 2007*; *Coates and Beynon, 2010*). To examine the role of SA-dependent defences, we profiled the expression of the SA-inducible marker gene *PR1* at 48 and 72 hr post-inoculation (hpi), which represents a critical time-window for host defence against *Hpa* (*Koch and Slusarenko, 1990*; *Soylu and Soylu, 2003*). None of the epiRILs showed a statistically significant increase in basal *PR1* expression after mock inoculation (*Figure 2a*; *Figure 1—figure supplement 4a*), indicating that the resistance is not based on constitutive up-regulation of SA-dependent defence signalling. However, in comparison to the Wt line, epiRILs #71, #148, #193, #229 and #508 showed augmented induction of *PR1* at 48 and/or 72 hpi with *Hpa* (*Figure 2a*; *Figure 1—figure supplement 4a*), indicating priming of SA-inducible defences (*Martinez-Medina et al., 2016*). To assess the role of cell wall defence, all lines were analysed for effectiveness of callose deposition, which is a pathogen-inducible defence mechanism that is largely controlled by SA-independent signalling (*Luna et al., 2011*). Compared to the Wt line, all but one epiRIL (#193) showed a statistically significant increase in the proportion of callose-arrested germ tubes (*Figure 2a*; *Figure 1—figure supplement 4b*). Hence, the eight most *Hpa*-resistant epiRILs are primed to activate differentially regulated defence responses, which explains the lack of major costs on growth and compatibility with other types of (a)biotic stress resistance in the epiRILs (*Figures 1b* and *2a*; *Figure 1—figure supplements 2–4*).

### Transgenerational stability of the resistance

The 123 epiRILs analysed for *Hpa* resistance had been self-pollinated for eight generations in a Wt (Col-0) genetic background since the F1 x Col-0 backcross (F9) (*Johannes et al., 2009*). To examine the transgenerational stability of the resistance phenotype over one more generation, five individuals from the eight most resistant epiRILs and the Wt line (*Figure 1a*, *Figure 1—figure supplement 3a*) were selected to generate F10 families, which were then tested for *Hpa* resistance. Comparing distributions of pooled leaves from all five families per line confirmed that each epiRIL maintained a statistically enhanced level of resistance (*Figure 1—figure supplement 5*; Pearson's Chi-squared test, p<0.05; top asterisks). However, when comparing individual F10 families to the Wt, 2 of the 40 F10 families (line #71–2 and line #148–2) exhibited Wt levels of susceptibility, indicating that they had lost *Hpa* resistance from the F9 to the F10 generation. Furthermore, four of the eight epiRILs tested (#71, #148, #545, and #508) displayed statistically significant variation in *Hpa* resistance between the 5 F10 families within the epiRIL (*Figure 1—figure supplement 5*; Pearson's Chi-squared test, p<0.05; † symbols), suggesting instability of the *Hpa* resistance.

### *Hpa*-resistant epiRILs show genome-wide priming of defence-related genes

To study the transcriptomic basis of the transgenerational resistance, Wt plants (line #602) and 4 *Hpa*-resistant epiRILs (#148, #193, #454 and #508), each carrying different combinations of the four epiQTLs, were analysed by RNA sequencing at 48 and 72 hpi (*Figure 2a*, bottom panel). Principal component analysis (PCA) of biologically replicated samples (n = 3) revealed clear separation between all treatment/time-point/epi-genotype combinations (*Figure 2b*). The first PCA axis explained 31% of the variation in transcript abundance, separating samples from mock- and *Hpa*-treated plants, whereas the second PCA axis explained 20% of the variation, mostly separating samples from the different lines (*Figure 2b*). This PCA pattern indicates that the response to *Hpa* infection had a bigger effect on global gene expression than epi-genotype. Moreover, samples from *Hpa*-inoculated epiRILs showed relatively little difference between both time-points (*Figure 2b*), whereas samples from *Hpa*-inoculated Wt plants at 48 hpi clustered between samples from mock-inoculated Wt plants and samples from *Hpa*-inoculated Wt plants at 72 hpi. This pattern suggests a difference in the speed and/or intensity of the transcriptional response to *Hpa*. To explore this possibility further, we performed three-factorial likelihood ratio tests ($q < 0.05$) to select differentially expressed genes between all epigenotype/treatment/time-point combinations. This analysis identified 20,569 genes, representing 61% of all annotated RNA-producing genes in the *Arabidopsis* genome, including transposable elements, non-coding RNA genes and pseudogenes

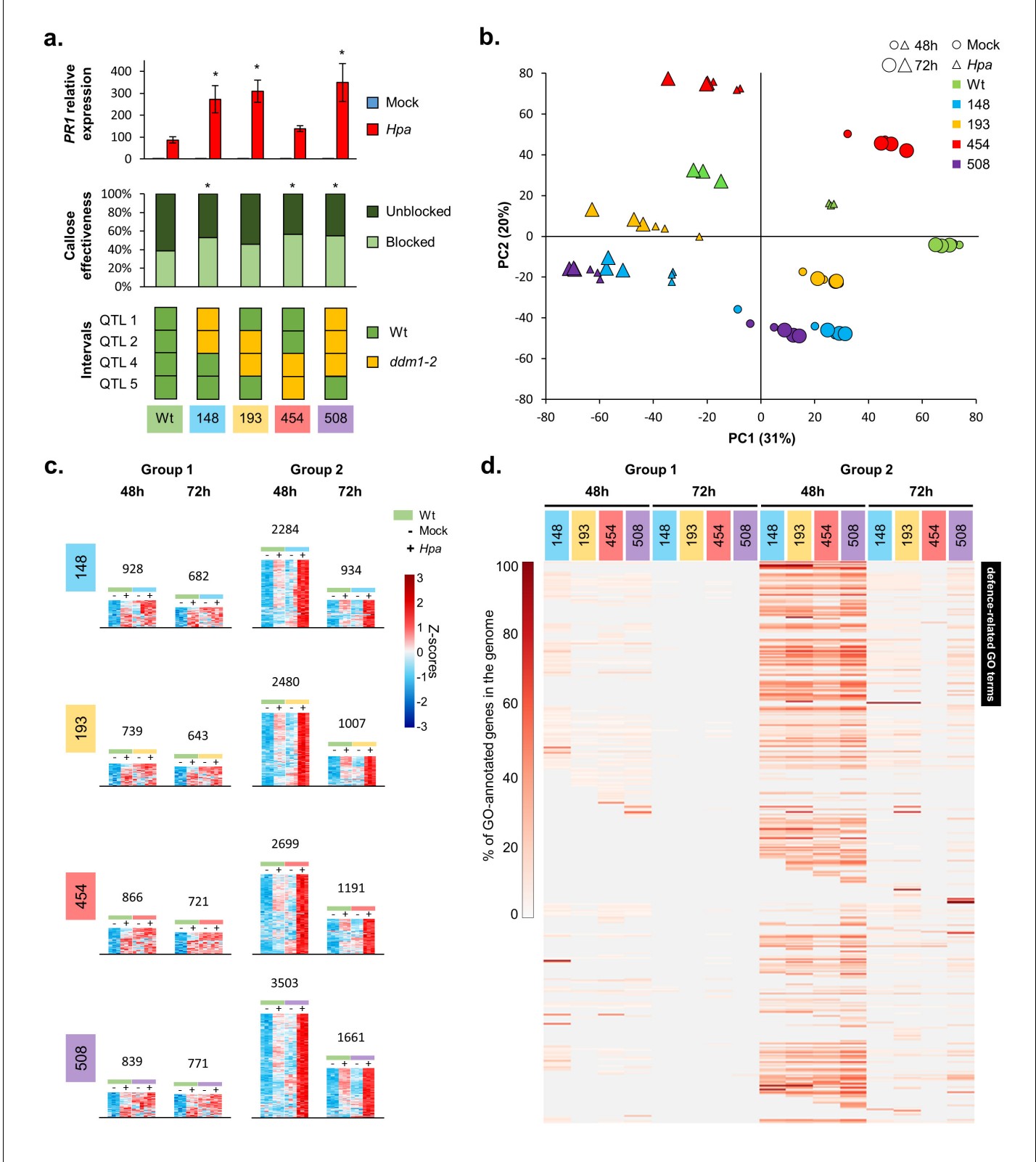

**Figure 2.** The defence-related transcriptome of *Hpa*-resistant epiRILs. (a) Defence marker phenotypes and epiQTL haplotypes of 4 *Hpa*-resistant epiRILs and the Wt (#602), which were analysed by RNA sequencing. Top graph: relative expression of SA-dependent *PR1* at 72 hr after inoculation (hpi) with *Hpa* (red) or water (blue). Middle graph: resistance efficiency of callose deposition in *Hpa*-inoculated plants. Shown are percentages of arrested (light) and non-arrested (dark) germ tubes at 48 hpi. Bottom panel: epiQTL haplotypes of selected lines. Green: methylated Wt haplotype; yellow:

*Figure 2 continued on next page*

*Figure 2 continued*

hypomethylated *ddm1-2* haplotype. Asterisks indicate statistically significant differences to the Wt. (see *Figure 1—figure supplement 4* for statistical information). (**b**) Principal component analysis of 27,641 genes at 48 (small symbols) and 72 (large symbols) hpi with *Hpa* (triangles) or water (Mock; circles). Colours indicate different lines. (**c**) Numbers and expression profiles of *Hpa*-inducible genes that show constitutively enhanced expression (Group 1) or augmented levels of *Hpa*-induced expression (Group 2) in the *Hpa*-resistant epiRILs at 48 or 72 hpi. Heatmaps show normalised standard deviations from the mean (z-scores) for each gene (rows), using *rlog*-transformed read counts (see *Figure 2—figure supplements 3* and *4* for better detail) (**d**) GO term enrichment of primed and constitutively up-regulated genes. Shown are 469 GO terms (rows), for which one or more epiRIL(s) displayed a statistically significant enrichment in one or more categories (Hypergeometric test, followed by Benjamini-Hochberg FDR correction; $q < 0.05$). Heatmap-projected values for each GO term (rows) represent percentage of GO-annotated genes in each category relative to all GO-annotated genes in the Arabidopsis genome (TAIR10). Black bar on the top right indicates 111 defence-related GO terms.

DOI: https://doi.org/10.7554/eLife.40655.014

The following source data and figure supplements are available for figure 2:

**Source data 1.**

DOI: https://doi.org/10.7554/eLife.40655.019

**Figure supplement 1.** Hierarchical clustering of differentially expressed genes (DEGs) in selected *Hpa*-resistant epiRILs and the Wt at 48 and 72 hpi (Ward method).

DOI: https://doi.org/10.7554/eLife.40655.015

**Figure supplement 2.** Selection of *Hpa*-inducible genes that show constitutively enhanced expression (Group 1) or enhanced *Hpa*-induced expression in the resistant epiRILs (Group 2).

DOI: https://doi.org/10.7554/eLife.40655.016

**Figure supplement 3.** Transcript profiles of *Hpa*-inducible genes showing constitutively enhanced expression in the *Hpa*-resistant epiRILs (Group 1).

DOI: https://doi.org/10.7554/eLife.40655.017

**Figure supplement 4.** Transcript profiles of *Hpa*-inducible genes showing enhanced levels of *Hpa*-induced expression in the *Hpa*-resistant epiRILs (Group 2).

DOI: https://doi.org/10.7554/eLife.40655.018

(*Supplementary file 1* dataset S3). Of these, 9364 genes were induced by *Hpa* at 48 and/or 72 hpi in one or more lines (*Supplementary file 1* dataset S4). Subsequent hierarchical clustering of this gene selection revealed a large cluster of *Hpa*-inducible transcripts displaying augmented induction in the epiRILs at the relatively early time-point of 48 hr after *Hpa* inoculation (*Figure 2—figure supplement 1*).

To characterize further the pathogen-inducible transcriptome of the resistant epiRILs, we selected *Hpa*-inducible genes showing elevated levels of expression in the epiRILs during *Hpa* infection. Within this gene selection, we distinguished two expression profiles. The first group of genes had been selected for constitutively enhanced expression in the resistant epiRILs, using the following criteria (Wald tests, $q < 0.05$): (*i*) *Hpa*-inducible in the Wt, (*ii*) not inducible by *Hpa* in the epiRIL and (*iii*) displaying enhanced accumulation in mock-treated epiRIL that is equal or higher than accumulation in the *Hpa*-inoculated Wt ('Group 1'; *Figure 2—figure supplement 2a*). The second group of genes had been selected for enhanced *Hpa*-induced expression in the epiRILs, using the following criteria (Wald tests, $q < 0.05$): (*i*) *Hpa*-inducible in the Wt (#602), (*ii*) *Hpa*-inducible in the epiRIL(s) and (*iii*) displaying statistically increased accumulation in *Hpa*-inoculated epiRILs compared to *Hpa*-inoculated Wt plants ('Group 2'; *Figure 2—figure supplement 2a*). For each epiRIL, we identified more genes in Group 2 than in Group 1 (*Figure 2c*; *Figure 2—figure supplements 2b*, *3* and *4*; *Supplementary file 1* datasets S5 and S6). This difference was most pronounced at 48 hpi, which represents a critical time-point for host defence against *Hpa* (*Koch and Slusarenko, 1990*; *Soylu and Soylu, 2003*). Analysis of a statistical interaction between epi-genotype x *Hpa* treatment revealed that >92% of all genes in Group 2 are significant for this interaction term (*Supplementary file 1* dataset S7), indicating a constitutively primed expression pattern. Visualisation of the expression profiles in heatmaps confirmed this notion, showing that the induction of Group 2 genes by *Hpa* is strongly augmented in the resistant epiRILs compared to the Wt line (*Figure 2c*; *Figure 2—figure supplement 4*), which is consistent with the definition of plant defence priming (*Martinez-Medina et al., 2016*).

To examine the functional contributions of the *Hpa*-inducible genes in Groups 1 and 2, we employed gene ontology (GO) term enrichment analysis. After exclusion of redundant GO terms (*Jantzen et al., 2011*), we identified 469 GO terms, for which one or more of the sets showed statistically significant enrichment. Group 2 genes at 48 hpi displayed dramatically enhanced GO term

enrichment compared to all other sets, which was obvious for all epiRILs (*Figure 2d*). This enrichment was particularly pronounced for 111 GO terms relating to SA-dependent and SA-independent defence mechanisms (*Supplementary file 1* dataset S8), which supports our phenotypic characterisation of SA-dependent and SA-independent defence markers (*Figure 1—figure supplement 4*). Collectively, these results suggest that the quantitative resistance of the epiRILs is based on priming of *Hpa*-inducible defence genes.

Interestingly, compared to the other gene selections, a relatively large proportion of defence-related genes in Group 2 at 48 hpi was shared between all four epiRILs (*Figure 2—figure supplement 2b*), pointing to relatively high similarity in the augmented immune response of the epiRILs. Furthermore, only 5% of the genes in the Group 1% and 6.5% of the genes in Group 2 are physically located within the borders of the epiQTL intervals (LOD drop-off = 2). The frequency of Group 1 and 2 genes relative to all other genes was significantly lower for the epiQTL regions compared to the entire *Arabidopsis* genome (14.6%; Pearson's Chi-squared test, p<0.05). Thus, the majority of *Hpa*-inducible Group 1 and 2 genes showing enhanced expression in the more resistant epiRILs are (*trans-*)regulated by DNA methylation at the four epiQTLs.

### The resistance epiQTLs do not contain defence genes that are *cis*-regulated by DNA methylation, suggesting involvement of *trans*-regulatory mechanisms

Although 92% of all genes in Group 2 were located outside the physical borders of the four epiQTL intervals (LOD-drop-off=2), we hypothesized that a small set of defence regulatory genes inside the epiQTL regions are directly (*cis*-)regulated by DNA methylation to mediate augmented levels of defence in response to *Hpa* infection. Since the Group 2 genes were strongly enriched with defence-related GO terms (*Figure 2d*), we examined whether their augmented expression during *Hpa* infection is associated with the hypomethylated *ddm1-2* haplotype. To this end, we calculated for each gene in Group 2 the ratio of normalized transcript abundance between *Hpa*-inoculated epiRIL and the *Hpa*-inoculated Wt line, which is proportional to their level of augmented expression during *Hpa* infection. Hierarchical clustering of these ratios enabled us to select for genes that exclusively show augmented expression when associated with the hypomethylated *ddm1-2* haplotype of the corresponding epiQTL (*Figure 3a*; *Figure 3—figure supplement 1a*). The expression ratios of 279 epiQTL-localised genes did not correlate with the *ddm1-2* haplotype (*Figure 3a*, cluster II; *Figure 3—figure supplement 1a*; *Supplementary file 1* dataset S9), indicating that DNA methylation does not *cis*-regulate their augmented *Hpa*-inducible expression. By contrast, 73 epiQTL-localised genes only showed augmented expression when associated with the hypomethylated *ddm1-2* haplotype (*Figure 3a*, cluster I; *Figure 3—figure supplement 1a*; *Supplementary file 1* dataset S10). To confirm the hypomethylated status of these genes, we performed comprehensive bisulfite sequencing analysis of DNA methylation for the four epiRILs and the Wt line. DMR analysis of the gene body (GB), 2 kb promoter region (P) and 1 kb downstream (D) regions confirmed that the levels of augmented gene expression of the 279 genes in cluster II do not correlate positively with the extent of DNA hypomethylation (*Figure 3b*, *Figure 3—figure supplement 1b*). This notion was confirmed by linear regression analysis between the augmented expression ratio (48 hpi) and the average level of DNA hypomethylation (*Figure 3—figure supplement 2*), indicating that the 279 genes in cluster II are regulated indirectly (in trans) by DNA methylation. By contrast, the 73 epiQTL-based genes in cluster I showed a positive correlation between augmented expression ratio (48 hpi) and DNA hypomethylation, which was statistically significant for each epiQTL (p<0.05; *Figure 3—figure supplement 2*). These results indicate that the 73 genes in cluster I are regulated locally (in cis) by DNA methylation.

Nearly all *cis*-regulated genes in cluster I showed a TE-like pattern of DNA methylation in the Wt (teM; methylation at CG, CHG and CHH contexts), whereas most cluster II genes showed either no methylation or a pattern of gene-body methylation in the Wt (gbM; methylation at CG only; *Figure 3b* and *Figure 3—figure supplement 1b*). Furthermore, dividing hypomethylation at gene bodies of Group 2 genes by type of DNA methylation (i.e. either teM or gbM) and plotting these values against augmented expression ratio revealed a statistically significant correlation between expression ratio and reduced teM (p=1,06e$^{-8}$; *Figure 3—figure supplement 3*), whereas no such correlation was found for reduced gbM (p=0.66; *Figure 3—figure supplement 3*). These results

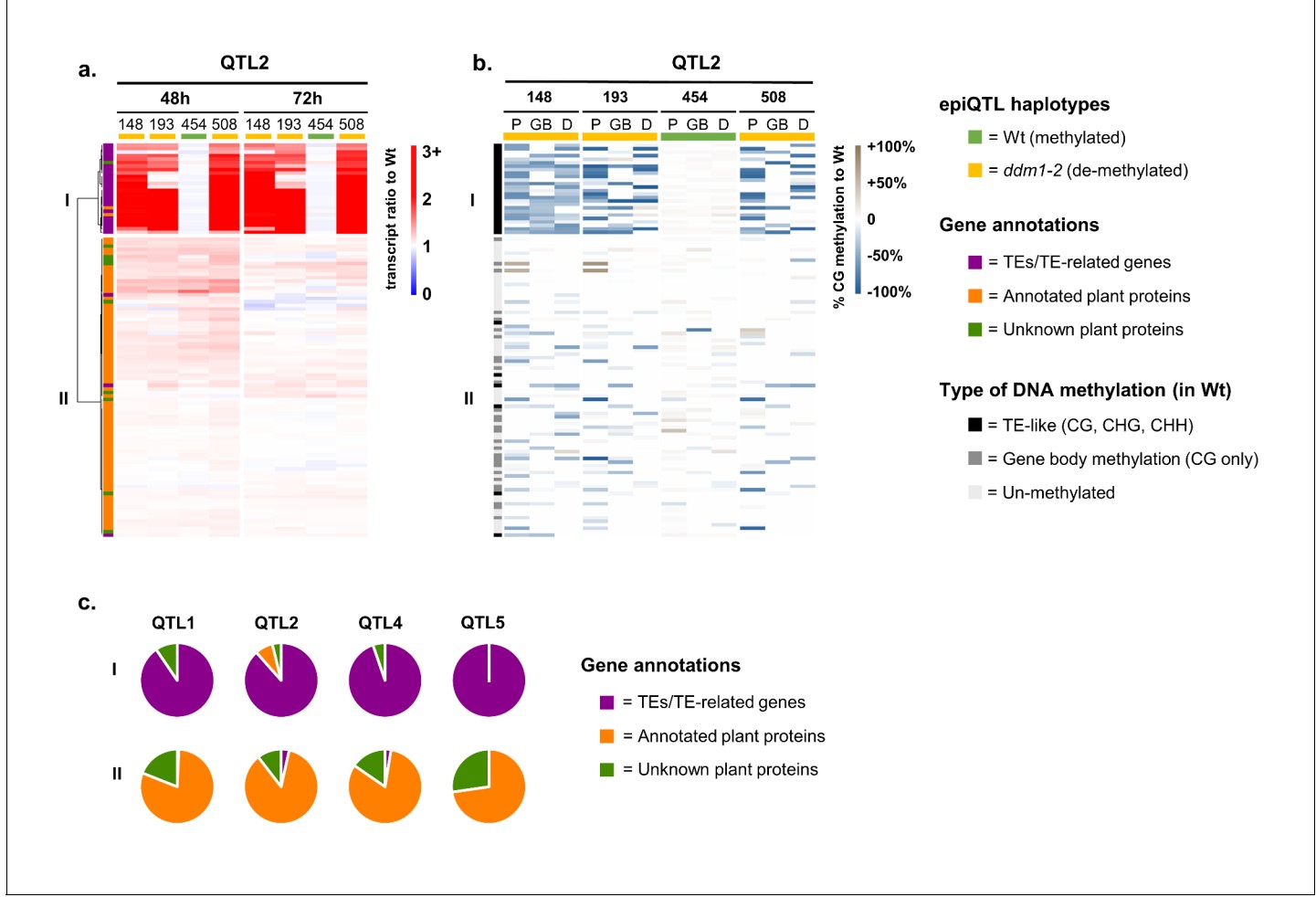

**Figure 3.** Relationship between augmentation of pathogen-induced expression and DNA methylation for epiQTL-localised genes. (a) Expression profiles of epiQTL-based genes showing elevated levels of *Hpa*-induced expression in one or more epiRIL(s) (Group 2). Shown are genes located in the epiQTL interval of chromosome II (epiQTL2; LOD drop-off = 2; see *Figure 3—figure supplement 1a* for other the epiQTLs). Heatmap shows gene expression ratios between *Hpa*-inoculated epiRILs and the Wt, representing augmented expression levels during pathogen attack. Hierarchical clustering yielded two distinctly regulated gene clusters (I and II). Coloured bars on the top indicate epiQTL2 haplotypes. Green: methylated Wt haplotype. Yellow: hypomethylated *ddm1-2* haplotype. (b) Levels of CG DNA methylation of the same genes in the epiQTL2 interval (see *Figure 3—figure supplement 1b* for other epiQTLs). Heatmap shows percentages of hypomethylation (blue) or hyper-methylation (brown) relative to the Wt for 2 kb promoter regions (P), gene bodies (GB) and 1 kb downstream regions (D). (c) Distribution of gene annotations of distinctly regulated gene clusters for each epiQTL.

DOI: https://doi.org/10.7554/eLife.40655.020

The following source data and figure supplements are available for figure 3:

**Source data 1.**
DOI: https://doi.org/10.7554/eLife.40655.026

**Figure supplement 1.** Relationship between augmentation of pathogen-induced expression and DNA methylation for epiQTL-localised genes.
DOI: https://doi.org/10.7554/eLife.40655.021

**Figure supplement 2.** Correlation analysis between augmented gene transcription and DNA hypomethylation.
DOI: https://doi.org/10.7554/eLife.40655.022

**Figure supplement 3.** Correlation analysis between augmented gene transcription and type of DNA hypomethylation.
DOI: https://doi.org/10.7554/eLife.40655.023

**Figure supplement 4.** Genomic contexts of six plant protein-encoding genes in the epiQTL intervals, whose transcriptional priming coincides with reduced DNA methylation.
DOI: https://doi.org/10.7554/eLife.40655.024

**Figure supplement 5.** Genome-wide chromatin interactions in Wt and *ddm1-2 Arabidopsis*.
DOI: https://doi.org/10.7554/eLife.40655.025

support the growing notion that reduced teM increases gene expression, whereas changes in gbM have no direct influence on gene expression (*Bewick et al., 2017*).

The majority of in cis-regulated genes in cluster I genes were annotated as TEs, such as DNA transposons of the *CACTA* family, retrotransposons of the *GYPSY* or *COPIA* families, or TE-related genes, encoding transposases or enzymes necessary for TE function (*Supplementary file 1* dataset S10). Only six genes were annotated as protein-coding genes, of which two shared homology to known protein-encoding genes (At2G07240, cysteine-type peptidase; At2G07750, RNA helicase). However, none of these two genes has previously been associated with plant defence. Furthermore, analysis of the genomic context of the six protein-coding genes revealed the presence of overlapping and/or nearby TEs (*Figure 3—figure supplement 4*), suggesting that their correlation between augmented expression and DNA hypomethylation is determined by association with TEs. Since TE-encoded proteins have no antimicrobial activity or direct defence regulatory function, our results suggest that global defence gene priming by hypomethylated epiQTLs is not based on *cis*-regulation of defence regulatory genes, but rather on alternative *trans*-acting mechanisms by DNA methylation of the TE-rich epiQTL.

## Discussion

By screening the Col-0 x *ddm1-2* epiRIL population for leaf colonisation by the downy mildew pathogen *Hpa*, we have identified four epiQTLs that provide quantitative disease resistance (*Figure 1b*). The combined contribution of all 4 DMR peak markers was estimated at 60% of the total variation (*Figure 1d*), which is higher than previously reported variation in developmental plant traits for this population (*Latzel et al., 2013*; *Cortijo et al., 2014*; *Kooke et al., 2015*). It was previously shown that half of all stably inherited DMRs in the Col-0 x *ddm1-2* epiRILs also occur in natural *Arabidopsis* accessions (*Latzel et al., 2013*; *Schmitz et al., 2013*). Considering that the epiRIL population includes heritable variation in a range of ecologically important plant traits, including flowering, root growth, nutrient plasticity and (a)biotic stress resistance (*Latzel et al., 2013*; *Cortijo et al., 2014*; *Kooke et al., 2015*), it is tempting to speculate that variation in DDM1-dependent DNA methylation contributes to natural variation and environmental adaptation of *Arabidopsis*. Indeed, the phenotypic diversity in the Col-0 x *ddm1-2* epiRIL population closely resembles that of natural *Arabidopsis* accessions (*Roux et al., 2011*; *Mauch-Mani et al., 2017*). Furthermore, independent studies have shown that high levels of enduring (a)biotic stress can trigger transgenerational acquired resistance (TAR) in Arabidopsis (*Luna et al., 2012*; *Wibowo et al., 2016*; *Rasmann et al., 2012*). Interestingly, repeated inoculation of 2- to 5 weeks old Arabidopsis seedlings with the hemi-biotrophic leaf pathogen *Pseudomonas syringae* pv. *tomato* causes TAR, which is associated with reduced transcription of *DDM1* gene in local leaves that is maintained in the apical meristem of paternal plants (Furci and Ton, unpublished results). To what extent this this prolonged repression in *DDM1* gene transcription causes heritable reduction in DNA methylation at the epiQTLs requires further study.

*Aller et al. (2018)* have recently used the same Col-0 x *ddm1-2* epiRIL population to map the contribution of heritable variation in DNA methylation to the production of defence-related glucosinolate metabolites (*Aller et al., 2018*). Interestingly, the resistance epiQTL on chromosome I from our study partially overlaps with an epiQTL that influences basal production of the aliphatic glucosinolate 3-methylthiopropyl (3MTP) (*Aller et al., 2018*). Glucosinolates contribute to defence against both herbivores and microbes (*Ishida et al., 2014*). Moreover, myrosinase-dependent breakdown products of indole-derived 4-methoxy-indol-3-ylmethylglucosinolate have been linked to the regulation of callose-mediated cell wall defence in Arabidopsis (*Clay et al., 2009*; *Bednarek et al., 2009*). However, the 3MTP-controlling epiQTL identified by *Aller et al. (2018)* was relatively weak compared to the epiQTL controlling *Hpa* resistance (*Figure 1b*), indicating that its contribution to *Hpa* resistance would at most be marginal. Furthermore, our transcriptome analysis revealed that the largest variation in gene expression between epiRILs and the Wt line comes from the transcriptional response to *Hpa*, rather than differences in basal gene expression (*Figure 2b–c*). Moreover, the genes in Group 2, which displayed enhanced *Hpa*-induced expression in the resistant epiRILs at the critical early time-point of 48 hpi, were strongly enriched with defence-related GO terms (*Figure 2d*). The majority of these Group 2 genes showed a statistically significant interaction between epi-genotype and *Hpa* treatment (*Supplementary file 1* dataset S7), indicating that these epiRILs were primed to activate defence-related genes. This notion was supported by the actual

expression profiles of Group 2 genes (*Figure 2c*; *Figure 2—figure supplement 4*), as well as the defence phenotypes of the eight most resistant epiRILs in the population (*Figure 2a*; *Figure 1—figure supplement 4*). Furthermore, our epiRIL screen for growth phenotypes demonstrated that the resistance-controlling epiQTLs do not have a major impacts on plant growth (*Figure 1b*), which is consistent with previous findings that defence priming is a low-cost defence strategy (*van Hulten et al., 2006*). While we cannot exclude other mechanisms, these independent lines of evidence collectively indicate that genome-wide priming of defence genes is the most plausible mechanism by which the epiQTLs mediate quantitative disease resistance in the population.

Over recent years, various studies have established a link between DNA hypomethylation and plant immune priming (*Espinas et al., 2016*; *Conrath et al., 2015*; *López Sánchez et al., 2016*). However, causal evidence that heritable regions of reduced DNA methylation mediate transgenerational disease resistance is lacking. Our study has shown that heritable regions of hypomethylated DNA are sufficient to mediate resistance in a genetic Wt background. Furthermore, our study is the first to link phenotypic and epigenetic variation of selected epiRILs to profiles of global gene expression, revealing that epigenetically controlled resistance is associated with genome-wide priming of defence-related genes (*Figure 2b–d*; *Figure 2—figure supplement 1*; *Figure 2—figure supplement 4*). The majority of these pathogenesis-related genes showed augmented induction at 48 hpi (*Figure 2c*), which represents a critical early time-point in the interaction between Arabidopsis and *Hpa*, during which hyphae from germinating spores start to penetrate the epidermal cell layer and invade the mesophyll (*Koch and Slusarenko, 1990*; *Soylu and Soylu, 2003*). Notably, this set of primed genes was substantially more enriched in SA-dependent and SA-independent defence GO terms than the set of *Hpa*-inducible genes that were constitutively up-regulated in *Hpa*-resistant epiRILs (*Figure 2d*), corroborating the analysis of phenotypical defence markers (*Figure 2a*; *Figure 1—figure supplement 4*).

DNA methylation of TEs has been reported to *cis*-regulate expression of nearby genes in *Arabidopsis* (*Soppe et al., 2000*; *Saze and Kakutani, 2007*; *Kinoshita et al., 2007*; *Lei et al., 2015*; *Williams et al., 2015*). By contrast, our study did not find evidence that DNA methylation in the epiQTLs *cis*-regulates the responsiveness of nearby of defence genes. Firstly, the majority of primed defence genes in the *Hpa*-resistant epiRILs were located outside the epiQTL intervals (92%). Secondly, of all primed genes within the epiQTLs, only 73 showed augmented induction that coincided with DNA hypomethylation (*Figure 3a*; *Figure 3—figure supplement 1*; *Figure 3—figure supplement 2*; *Supplementary file 1* dataset S10). Of these, 67 encoded TEs or TE-related genes, while the six protein-encoding genes were closely associated with one or more TEs and did not have functions related plant defence (*Figure 3a*; *Figure 3—figure supplement 1*; *Supplementary file 1* dataset S10). Since TEs do not encode defence signalling proteins, we propose that DNA hypomethylation at the TE-rich epiQTLs mediates augmented induction of defence genes across the genome via *trans*-acting mechanisms. A recent transcriptome study of *Hpa*-infected *Arabidopsis* identified 166 defence-related genes that were primed in the hypomethylated *nrpe1-11* mutant and/or repressed in hyper-methylated *ros1-4* mutant (*López Sánchez et al., 2016*). The majority of these defence genes were not targeted by NRPE1- and/or ROS1-dependent DNA (de)methylation, indicating that their responsiveness is *trans*-regulated by DNA methylation. Although NRPE1 and ROS1 target partially different genomic loci than DDM1[20], this study supports our hypothesis that DNA methylation controls global defence gene responsiveness via *trans*-acting mechanisms.

There are various mechanisms by which DNA methylation could *trans*-regulate defence gene expression. It is possible that transcribed TEs in the hypomethylated epiQTLs generate 21-22nt or 24nt small RNAs (sRNAs) that influence distant heterochromatin formation through via RDR6- and DCL3-dependent RdDM pathways (*Panda et al., 2016*). Support for *trans*-regulation by sRNAs came from a recent study, which reported that induction and subsequent re-silencing of pericentromeric TEs in Arabidopsis upon *Pseudomonas syringae* infection is accompanied with accumulation of RdDM-related sRNAs that are complementary to TEs and distal defence genes. Interestingly, while the accumulation of these sRNAs coincided with re-silencing of the complementary TEs, the complementary defence genes remained expressed in the infected tissues (*Cambiagno et al., 2018*). These findings are supported by another recent study, which demonstrated that AGO1-associated small RNAs can *trans*-activate distant defence gene expression through interaction with the SWI/SNF chromatin remodelling complex (*Liu et al., 2018*). Apart from sRNAs, it is also possible that long intergenic noncoding RNAs (lincRNAs) from the hypomethylated epiQTLs regulate pathogen-induced

expression of distant defence genes. A recent study revealed that pericentromeric TEs of *Arabidopsis* can produce DDM1-dependent lincRNAs that are increased by abiotic stress exposure (*Wang et al., 2017a*). Since lincRNAs can promote euchromatin and heterochromatin formation at distant genomic loci (*Heo et al., 2013*; *Quinodoz and Guttman, 2014*), hypomethylated TEs within the epiQTLs could generate priming-inducing lincRNAs. While knowledge about lincRNAs in plants remains limited, like sRNAs, their activity depends on sequence complementary with target loci (*Vance and Ponting, 2014*). Unlike non-coding RNAs, long-range chromatin interactions can *trans*-regulate gene expression independently of sequence complementarity (*Harmston and Lenhard, 2013*; *Liu, 2016*; *Weber et al., 2016*; *Wang et al., 2017b*). Previous high-throughput chromosome conformation capture (Hi-C) analysis revealed that the *ddm1-2* mutation has a profound impact on long-range chromatin interactions within and beyond the pericentromeric regions (*Feng et al., 2014*). Projection of these DDM1-dependent interactions onto the *Arabidopsis* genome shows extensive coverage of the resistance epiQTLs identified in this study (*Figure 3—figure supplement 5*). Whether these long-range interactions contribute to *trans*-regulation of defence gene priming would require further study, including a fully replicated Hi-C analysis of the resistant epiRILs characterised in this study.

In conclusion, our study has shown that heritable DNA hypomethylation at selected pericentromeric regions controls quantitative disease resistance in *Arabidopsis*, which is associated with genome-wide priming of defence-related genes. This transgenerational resistance is not associated with reductions in plant growth (*Figure 1b*), nor does it negatively affect resistance to other types of (a)biotic stresses tested in this study (*Figure 1—figure supplement 3*). However, whether this form of epigenetically controlled resistance can be exploited in crops depends on a variety of factors, including the stability of the disease resistance and potential non-target effects. For instance, our experiments with Arabidopsis revealed that the resistance has limited stability and can erode over one more generation in some epiRILs (*Figure 1—figure supplement 5*). Furthermore, the genomes of most crop species contain substantially higher numbers of TEs, rendering predictions about the applicability and potentially undesirable side effects on growth and seed production uncertain. Future research will have to point out whether introgression of hypomethylated pericentromeric loci into the background of elite crop varieties allows for selection of meta-stable quantitative disease resistance without side-effects on agronomically important traits.

## Materials and methods

### Plant material and growth conditions

Epigenetic recombinant inbred lines (epiRILs) seeds of *Arabidopsis* (*Arabidopsis thaliana*, accession Col-0) were purchased from Versailles Arabidopsis Stock Centre, INRA, France (http://publiclines.versailles.inra.fr/epirils/index). The epiRIL screen included siblings of the F4 *ddm1-2* parental plant of the epiRIL population (IBENS, France). *Arabidopsis* seeds were stratified in water at 4°C in the dark for 3-5 days. For pathogen bioassays, seeds were sown in a sand:compost mixture (1:3) and grown at short-day conditions for 3 weeks (8.5 hr light/15.5 hr dark, 21°C, 80% relative humidity,~125 μmol s$^{-1}$ m$^{-1}$ light intensity). To test transgenerational inheritance and stability of *Hpa* resistance in the eight most resistant epiRILs (*Figure 1—figure supplement 5*), five individual F9 plants were cultivated for 4 weeks at short-day conditions and then moved to long-day conditions to initiate flowering (16 hr light/8 hr dark, 21°C, 80% relative humidity,~125 μmol s$^{-1}$ m$^{-1}$ light intensity). Seeds of the 40 F10 families were collected for analysis of *Hpa* resistance (see below).

### Screen for variation in disease resistance and seedling growth

Three-week-old seedlings were spray-inoculated with a suspension of asexual conidia from *Hyaloperonospora arabidopsidis* strain WACO9 (*Hpa*) at a density of $10^5$ spores/ml. *Hpa* colonizsation was quantified at 6 days post-inoculation (dpi) by microscopic scoring of leaves, as described previously (*López Sánchez et al., 2016*). Briefly, trypan blue-stained leaves were analysed with a stereomicroscope (LAB-30, Optika Microscopes) and assigned to 4 *Hpa* colonisation classes: class I, no hyphal colonisation; class II, ≤50% leaf area colonized by pathogen hyphae without formation of conidiophores; class III,≤75% leaf area colonized by hyphae, presence of conidiophores; class IV, >75% leaf area colonized by the pathogen, abundant conidiophores and sexual oospores (*Figure 1—figure*

*supplement 1*). At least 100 leaves per (epi)genotype were analysed, not including the cotyledons. Statistically significant differences in frequency distribution of *Hpa* colonisation classes between lines were determined by Pearson's Chi-squared tests, using R (v.3.5.1). Growth analysis of the epiRIL population was based on digital photos (Canon 500D, 15MP) of 3-week-old plants, which were taken on the day of *Hpa* inoculation. Digital image analysis of total green leaf area (GLA) was performed using Adobe Photoshop 6.0. Green pixels corresponding to GLA were selected and converted into mm$^2$ after colour range adjustment, using the magic wand tool.

## Mapping of epigenetic quantitative trait loci (epiQTLs)

Mapping of epiQTLs was performed using the '*scanone*' function of the R/qtl package for R (*Broman et al., 2003*) (Haley-Knott regression, step size: 2 cM), combining experimental phenotypical data with the recombination map of differentially methylated regions (DMR) generated previously (*Colomé-Tatché et al., 2012*). For analysis of *Hpa* resistance, the categorical scoring of *Hpa* resistance was first converted into a numeric resistance index (RI), using the following formula:

$$RI = (f_{\text{class I}} {}^\star 4) + (f_{\text{class II}} {}^\star 3) + (f_{\text{class III}} {}^\star 2) + (f_{\text{class IV}} {}^\star 1)$$

where *f* = relative frequency of *Hpa* colonisation class of each line, multiplied by an arbitrary weight value ranging from four for the most resistant category (class I) to one for most susceptible category (class IV). Mapping of epiQTLs controlling plant growth was based on average GLA values of each line before *Hpa* infection. A logarithm of odds (LOD) threshold of significance for each trait was determined on the basis of 1000 permutations for each dataset (α = 0.05). The proportion of phenotypic variance $R^2$ *(G)* explained by the DMR markers with the highest LOD score (peak markers) of all four epiQTLs was calculated with the following formula (*Cortijo et al., 2014*):

$$R^2(G) = 1 - \frac{n-1}{n-(k+1)} \frac{\sum_i^n \left( y_i - \left[ \beta_0 + \sum_j^k \beta_j g_{ij} \right] \right)^2}{\sum_i^n (y_i - \bar{y})^2}$$

where *n* = number of lines analysed, *k* = number of DMR markers tested; $\beta_0$ = intercept of the multiple regression model; $\beta_j$ = QTL effect for each QTL *j* (slopes for each marker in the multiple regression model); $g_{ij}$ = (epi) genotype of the $j^{\text{th}}$ marker for each individual *i* (coded as '1' for *ddm1-2* epialleles and '-1' for WT epialleles); $y_i$ = phenotypic value of individual *i*; $\bar{y}$ = mean of phenotypic values. The contribution of each individual QTL *j* ($R^2$ *(g)*) was calculated, using the following formula:

$$R^2(g) = 1 - \frac{n-1}{n-(k+1)} \frac{\beta_j^2 \sum_i^n (g_{ij} - \bar{g}_j)^2}{\sum_i^n (y_i - \bar{y})^2} \quad ,$$

as described by (*Cortijo et al., 2014*), where *n* = number of lines analysed, *k* = number of markers tested; $\beta_j$ = QTL effect for each QTL *j* (slopes for each peak marker in the multiple regression model); $g_{ij}$ = (epi)genotype of the $j^{th}$ marker for each individual *i* (coded as '1' for *ddm1-2* epialleles and '-1' for WT epialleles); $\bar{g}_j$ = average of the (epi)genotypes values for the $j^{th}$ marker. Covariance was calculated by subtracting the sum of the individual contributions of each QTL *j* on phenotypical variance (i.e. $R^2(g_{QTL1})$ + $R^2(g_{QTL2})$ + $R^2(g_{QTL4})$ + $R^2(g_{QTL5})$) from the phenotypical variance explained by the full model (i.e. $R^2(G)$).

## Analysis of shared transposition events

TE-tracker software was used to interrogate available Illumina whole-genome sequencing data from 122 epiRILs for the presence of >2 shared transposition evens (STEs) within the epiQTLs intervals (*Gilly et al., 2014*). STEs were analysed for statistically significant linkage with resistance phenotypes (RIs), using the same linear regression model as described above for DMR linkage analysis.

## *Plectosphaerella cucumerina* pathoassays

*Plectosphaerella cucumerina* (*Pc*, strain BMM (*Ton and Mauch-Mani, 2004*)) was grown from frozen agar plugs (−80° C) on potato dextrose agar (PDA; Difco, UK). Inoculated plates were maintained at room temperature in the dark for at least 2 weeks. Spores were gently scraped from water-

inundated plates, after which spore densities were adjusted to $10^6$ spores/ml using a hemocytometer (Improved Neubauer, Hawksley, UK). Four fully expanded leaves of similar age from 5-week-old plants were inoculated by applying 5 µl droplets, minimising variability due to age-related resistance. After inoculation, plants were kept at 100% RH until scoring of lesion diameters. Average lesion diameters at nine dpi were based on four leaves per plant from 12 plants per (epi)genotype ($n = 40$–48), using a precision caliper (Traceable, Fischer Scientific). Statistically significant differences in necrotic lesions diameter (asterisks) were quantified by two-tailed Student's t-test ($p < 0.05$) in pairwise comparisons with Wt line (#602), using R (v3.5.1).

## Salt stress tolerance assays

Seeds were sterilised by exposure for 4 hr (h) to chlorine vapours from a 200 ml bleach solution containing 10% v/v hydrochloric acid (37% v/v HCl, Fischer Scientific, 7732-18-5). Seeds were air-dried for 1 hr in a sterile laminar flow cabinet and plated on half strength MS plates (Duchefa, M0221; +0.05% w/v MES,+1% w/v sucrose, pH 5.7), containing increasing concentrations of NaCl (0 mM, 50 mM, 75 mM and 100 mM; Fischer Scientific, 7647-14-5). Plates were stratified for 4 days in the dark at 4°C and transferred to short-day growth conditions (8.5 hr light/15.5 hr dark, 21°C, 80% RH, light intensity 100–140 µmol $s^{-1}$ $m^{-1}$). Salt tolerance was expressed as percentage of seeds producing fully expanded cotyledons by 6 days after stratification. Germination percentages of epi-genotypes were calculated from >50 seeds per treatment. Statistically significant differences in germination rates (asterisks) were quantified by Fisher's exact test ($p < 0.05$) in pairwise comparisons with Wt line (#602) at each salt concentration, using R (v3.5.1).

## Quantification of callose effectiveness against *Hpa* infection

Seedlings were collected at three dpi and cleared for >24 hr in 100% ethanol. One day prior to analysis, samples were incubated for 30 min in 0.07 M phosphate buffer (pH 9), followed by 15 min incubation in a 4:1 mixture (v/v) of 0.05% w/v aniline blue (Sigma-Aldrich, 415049) in 0.07M phosphate buffer (pH 9) and 0.025% w/v calcofluor white (Fluorescent brightener 28, Sigma-Aldrich, F3543) in 0.1M Tris-HCL (pH 7.5). After initial staining, samples were incubated overnight in 0.5% w/v aniline blue (Sigma-Aldrich, 415049) in 0.07M phosphate buffer (pH 9) and scored with an epifluorescence microscope (Olympus BX 51) fitted with blue filter (XF02-2; excitation 330 nm, emission 400 nm). Germinated conidia (germ tubes) were divided between in two classes: non-arrested and arrested by callose. In each assay, 10 leaves from different plants for each (epi)genotype were analysed, amounting to >150 conidia-callose interactions. Statistically significant differences in resistance efficiency of callose (asterisks) were analysed using Pearson's Chi-squared tests ($p < 0.05$) in pairwise comparisons with Wt line (#602), using R (v3.5.1).

## Reverse-transcriptase quantitative polymerase chain reactions (RT-qPCR)

Three biologically replicated samples for each genotype/treatment/time-point combination were collected at 48 and 72 hpi, each consisting of six to 12 leaves collected from different plants per pot. Samples were snap-frozen in liquid nitrogen and ground to a fine powder, using a tissue lyser (QIAGEN TissueLyser). Total RNA was extracted using a guanidinium thiocyanate-phenol-chloroform extraction isolation protocol. Frozen powder was vortexed for 30 s in 1 ml Extraction buffer: 1M guanidine thiocyanate (Amresco, 0380), 1M ammonium thiocyanate (Sigma-Aldrich, 1762-95-4), 0.1M sodium acetate (Fisher Scientific, 127-09-3), 38% v/v AquaPhenol (MP Biomedicals, 108-95-2) and 5% v/v glycerol (Fisher Scientific, 56-81-5). Samples were incubated at room temperature (RT) for 1 min and then centrifuged for 5 min at 16,500 *g*. The supernatant was then transferred to a new tube, mixed with 200 µl chloroform and vortexed for 10–15 s. After centrifuging for 5 min (16,500 *g*), the aqueous phase was transferred to new tubes, gently mixed by inversion with 350 µl 0.8M sodium citrate (Sigma-Aldrich, 6132-04-3) and 350 µl isopropanol (Fischer Chemicals, 67-63-0) and left at RT for 10 min for RNA precipitation. Samples were centrifuged for 15 min at 16,500 *g* (4°C), after which pellets were washed twice in 1 ml 70% ethanol, centrifuged at 16,500 *g* for 1 min, and air-dried before dissolving in 50 µl nuclease-free water. Total RNA was quantified, using a Nanodrop 8000 Spectrophotometer (Thermo Scientific). RNA extracts were treated with DNaseI, using the RQ1 RNase-Free DNase kit (Promega, M6101). First-strand cDNA synthesis was performed from 1 µg

RNA, using SuperScript III Reverse Transcriptase (Invitrogen, 18080093) according to the supplier's recommendations. The qPCR reactions were carried out with a Rotor-Gene Q real-time PCR cycler (Qiagen) and the Rotor-Gene SYBR Green PCR Kit (Qiagen, 204074). Relative *PR1* gene expression was calculated, using Livak's ΔΔCT method (*Livak and Schmittgen, 2001*) with correction for average PCR efficiencies for each primer pair across experiment samples. Gene expression was normalised against average expression values of At1G13440 (GAPDH), At5G25760 (UBC) and At2G28390 (SAND family protein) (*Czechowski et al., 2005*). Reactions were performed using previously described primer sequences (*López Sánchez et al., 2016*). Statistically significant differences in relative expression (asterisks) were quantified by two-tailed Student's t-test (p<0.05) in pairwise comparisons with *Hpa*-treated Wt line (#602).

## Transcriptome analysis

Samples for RNA sequencing were collected at 48 and 72 hpi of 3-week-old plants. Every epi-genotype/treatment/time-point combination was based on three biologically replicated samples, each consisting of 6–12 shoots from different plants. Initial RNA extraction was performed as described for RT-qPCR reactions. Prior to library preparation, RNA concentration and integrity were measured, using 2100 Bioanalyzer (Agilent) with provided reagents kits and according to manufacturer's instructions. All RNA samples had RNA integrity numbers (RIN) >7.5. Sequencing libraries were prepared from total RNA, using the TruSeq Stranded Total RNA kit and Ribo-Zero Plant leaf kit (Illumina, RS-122–2401), according to the manufacturer's instructions. Sequencing runs were performed on a HiSeq1500 platform (Illumina), generating paired-end reads of 125 bp and an average quality score (Q30) >93%. Each sample generated around 35 million paired reads.

Read quality was assessed by FastQC software (*Andrews, 2010*). Read length and distribution were optimized and adapter sequences were trimmed, using Trimmomatic software (*Bolger et al., 2014*). Reads were aligned and mapped to the *Arabidopsis* genome (TAIR10 annotation), using splice site-guided HISAT2 alignment software (John Hopkins University, second iteration of (*Kim et al., 2015*)). For all samples, more than 95% of reads could successfully be mapped once or more onto the *Arabidopsis* genome. Number of reads per gene were quantified with the Python package *HTseq* (*Anders et al., 2015*). Differential expression analysis was performed using the *DESeq2* R package, which applies a negative binomial generalized linear model to estimate mean and dispersion of gene read counts from the average expression strength between samples (*Love et al., 2014*). Prior to principal component analysis (PCA) by the *plotPCA* function, gene read counts were subjected to regularized logarithmic transformation, using the *rlog* function (*Love et al., 2014*). Likelihood ratio tests of variance within a three-factorial linear model for epigenotype, treatment, time-point and interactions thereof were used to identify genes showing differences in expression across one or more factors (*Love et al., 2014*). Differentially expressed genes (DEGs) were subjected to hierarchical clustering (Ward method) and presented as a heat map, using the *pheatmap* R package (*Kolde, 2015*). For each gene, *rlog*-normalized read counts of each sample were subtracted from the mean of all samples, and divided by the standard deviation to facilitate heatmap visualisation (z-score). To identify DEGs between two treatment/time-point/epi-genotype combinations, pair-wise comparisons (Wald test; q < 0.05) were performed with the DEGs selection obtained by the *lrt* test, using the selection criteria illustrated in *Figure 2—figure supplement 2a*. All *Hpa*-inducible genes in the Wt and/or epiRILs were selected for elevated expression in the more resistant epiRILs during *Hpa* infection. Subsequently, these genes were divided between two groups based on their expression profile. Group 1 genes were selected for constitutively enhanced expression in the epiRIL(s) relative to the Wt (*Figure 2—figure supplements 2* and *3*); Group 2 genes were selected for enhanced levels of *Hpa*-induced expression in the epiRIL(s) relative to the Wt (*Figure 2—figure supplements 2* and *4*). To determine the number of Group 2 genes that show a statistically significant interaction between epigenotype x *Hpa* treatment (all 16,009 genes significant for this interaction were selected from the three-factorial linear model, using the *contrast* function, and cross-referenced against Group 2 genes.

Gene ontology (GO) term enrichment analysis was performed with the Plant GSEA toolkit (*Yi et al., 2013*). GO terms were checked for significant enrichment against the whole genome background, using a hypergeometric test and Benjamini-Hochberg false discovery rate correction (q < 0.05). Lists of enriched GO terms in each treatment were analysed by the GO Trimming 2.0 algorithm (*Jantzen et al., 2011*) to remove redundancy of terms, applying a soft trimming threshold

of 0.40. The output list from GO Trimming 2.0 was run through GOSlim Viewer (AgBase) to reduce GO terms according to GO slim ontologies (GO consortium). Enrichment was quantified as the percentage of GO term-annotated genes within a certain selection relative to the total number of *Arabidopsis* genes in that GO term.

## Methylome analysis

For each line, three independent biological replicates were collected, consisting of pooled leaves from six plants of the same developmental stage. High-quality genomic DNA was extracted from leaves of 5-week-old plants, using the GenElute Plant Genomic DNA Miniprep Kit (Sigma-Aldrich). Bisulfite sequencing was performed by GATC Biotech (UK). After quality trimming of read sequences, adapter sequences were removed, and reads were filtered by Cutadapt (version 1.9; Pair end-mode; phred score = 20, min.length = 40). Reads were mapped to an index genome, using of BS-Seeker2 (version 2.0.10, mismatch = 0.05, maximum insert size = 1000 bp). Bowtie2 (version 2.2.2) was used for alignment of reads, as described previously (*Lauss et al., 2018*). Differential methylation for promoter regions (−2 kb), gene bodies, and downstream regions (+1 kb) relative to the Wt was called using methylkit (version 1.0.0; minimum coverage = 5 x, $q$ = 0.05). Differentially methylated states were visualised as a heat map, using the '*pheatmap*' R package (version 1.0.8) (*Kolde, 2015*).

To differentiate Wt methylation states of all epiQTL-based genes in Group 2 (see above), gene bodies of all nuclear genes were categorised between un-methylated, gene body methylated (gbM; CG context only) or TE-like methylated (teM; CHG and/or CHH with or without CG). For each gene containing 20 or more cytosines, methylated and un-methylated cytosine base calls in each context were extracted from the sequence read alignments. Positions with less than 4x coverage were ignored. Methylation patterns were categorised as TE-like if methylated read calls relative to un-methylated read calls in CHG and/or CHH contexts showed a statistically significant increase over average methylation rates of all genes across the genome in the respective context, using the '*binom.test*' function in R (FDR-adjusted p<0.01). The remaining genes were classified either as gbM if the same test revealed a statistically significant increase in CG context, or as un-methylated if no statistically significant increase in DNA methylation could be detected in any sequence context.

## Correlation analysis between gene expression and DNA methylation

Correlations between augmented expression ratio of Group 2 genes (see Transcriptome analysis) and DNA hypomethylation (CG), were determined by plotting augmented gene ratios at 48 hpi against average hypomethylation compared to Wt (%) across promoter region, gene body, and downstream region (see Methylome analysis). To determine which type of DNA hypomethylation correlates with augmented expression in the epiRILs, hypomethylation at gene bodies of Group 2 genes were divided between teM and gbM and plotted against the corresponding expression ratios at 48 hpi. If hypomethylation occurred at CG context only, genes were classified as being reduced in gene body methylation (gbM); if hypomethylation occurred all three sequence contexts (CG, CHG, CHH), genes were classified as being reduced in TE methylation (teM). Values of gbM hypomethylation were expressed as percentage reduction in GC methylation relative to the Wt; values of teM hypomethylation were expressed as percentage reduction in all sequence contexts. Linear regression analyses were performed using R software (v.3.5.1).

## Hi-C analysis

HiC sequence libraries SRR1504819 and SRR1504824[64] were downloaded from NCBI SRA. Sequences were pre-processed and aligned to the TAIR10 Arabidopsis nuclear genome sequence (*Berardini et al., 2015*), using HiCUP (0.5.9) (*Wingett et al., 2015*) and Bowtie2 (*Langmead and Salzberg, 2012*) (2.2.6). Alignments were filtered and de-duplicated as part of the processing by HiCUP, before being further processed in HOMER (*Heinz et al., 2010*) (4.9.1) at 5 kb resolution. Differential interactions were assessed reciprocally, using each sample as background (analyzeHiC-ped). Interactions were determined to be potentially dependent on genotype if the absolute *z*-score of the primary versus the secondary experiment was more than 1. Visualisations were generated using Circos (*Krzywinski et al., 2009*)(0.69–5), based on bundled links (-max_gap 10001).

## Data availability

Transcriptome sequencing and bisulfite sequencing reads are available from the European Nucleotide Archive (ENA) under accession code PRJEB26953.

## Acknowledgements

We thank David Pardo and Ana Lopez for technical assistance in the lab. We thank the La Trobe University's Genomics Platform for technical support. The research was supported by a consolidator grant from the European Research Council (ERC; no. 309944 'Prime-A-Plant') to JT, a Research Leadership Award from the Leverhulme Trust (no. RL-2012–042) to JT and two BBSRC grants to JT (BB/L008939/1 and BB/P006698/1). Work in the group of VC was supported by the Agence Nationale de la Recherche (ANR-09-BLAN-0237 EPIMOBILE). FJ acknowledges support from the Technical University of Munich - Institute for Advanced Study funded by the German Excellent Initiative and the European Seventh Framework Programme under grant agreement no. 291763. FJ is also supported by the SFB/Sonderforschungsbereich924 of the Deutsche Forschungsgemeinschaft (DFG).

## Additional information

### Funding

| Funder | Grant reference number | Author |
|---|---|---|
| Agence Nationale de la Recherche | ANR-09-BLAN-0237EPIMOBILE | Vincent Colot |
| European Commission Seventh Framework Programme | 291763 | Frank Johannes |
| Deutsche Forschungsgemeinschaft | SFB924 | Frank Johannes |
| Leverhulme Trust | RL-2012-042 | Jurriaan Ton |
| H2020 European Research Council | 309944 | Jurriaan Ton |
| Biotechnology and Biological Sciences Research Council | BB/L008939/1 | Jurriaan Ton |
| Biotechnology and Biological Sciences Research Council | BB/P006698/1 | Jurriaan Ton |

The funders had no role in study design, data collection and interpretation, or the decision to submit the work for publication.

### Author contributions

Leonardo Furci, Data curation, Formal analysis, Investigation, Visualization, Methodology, Writing—original draft, Writing—review and editing, Performed the experiments, Wrote the manuscript; Ritushree Jain, Data curation, Software, Formal analysis, Visualization, Methodology, Writing—original draft, Writing—review and editing, Coordinated and performed data analysis of RNA-seq and bisulfite-seq experiments, Wrote the manuscript; Joost Stassen, Data curation, Software, Formal analysis, Methodology, Analysed Hi-C data and helped in analysing RNA-seq and bisulfite-seq experiments; Oliver Berkowitz, Investigation, Prepared RNA-seq libraries and conducted the sequencing; James Whelan, Supervision, Investigation, Prepared RNA-seq libraries and conducted the sequencing; David Roquis, Software, Formal analysis, Contributed to data analysis of bisulfite-seq experiment; Victoire Baillet, Software, Formal analysis, Performed genomic analysis of shared TE insertion events in the epiQTL intervals; Vincent Colot, Resources, Software, Formal analysis, Supervision, Performed genomic analysis of shared TE insertion events in the epiQTL intervals; Frank Johannes, Software, Formal analysis, Supervision, Contributed to data analysis of bisulfite-seq experiment; Jurriaan Ton, Conceptualization, Resources, Formal analysis, Supervision, Funding acquisition, Validation, Investigation, Visualization, Methodology, Writing—original draft, Project administration, Writing—review and editing, Conceived and supervised the project, Wrote the draft and revised manuscript

## Author ORCIDs

Leonardo Furci [iD] http://orcid.org/0000-0003-2155-6289
Joost Stassen [iD] https://orcid.org/0000-0001-5483-325X
Oliver Berkowitz [iD] https://orcid.org/0000-0002-7671-6983
Vincent Colot [iD] https://orcid.org/0000-0002-6382-1610
Jurriaan Ton [iD] http://orcid.org/0000-0002-8512-2802

## Decision letter and Author response

Decision letter https://doi.org/10.7554/eLife.40655.034
Author response https://doi.org/10.7554/eLife.40655.035

---

# Additional files

### Supplementary files

• Supplementary file 1. Supplementary datasets. (S1) Distribution of 126 DMR markers for Wt and *ddm1-2* haplotypes in all 123 epiRILs. (S2) Linkage between shared transposition events (STEs) in the epiQTL of Chromosome 1 and *Hpa* resistance. (S3) Differentially expressed genes between all epi-RIL/treatment/time-point combinations. (S4) Genes showing statistically significant induction by *Hpa* in the Wt and/or *Hpa*-resistant epiRILs. (S5) *Hpa*-inducible genes displaying constitutively up-regulated expression in one or more *Hpa*-resistant epiRILs. (Group 1). (S6) *Hpa*-inducible genes displaying augmented induction in one or more *Hpa*-resistant epiRILs. (Group 2). (S7) Percentages of genes in Group 2 showing statistically signigicant interaction effect between epigenotype and treatment. (S8) Defence-related GO terms showing significant enrichment in Group 1 and/or Group 2 genes. (S9) *Hpa*-inducible genes whithin epiQTL intervals that do not show positive correlation between augmented induction and DNA hypo-methylation (*trans*-regulated). (S10) *Hpa*-inducible genes whithin epiQTL intervals that show positive correlation between augmented induction and DNA hypo-methylation (*cis*-regulated).
DOI: https://doi.org/10.7554/eLife.40655.027

• Transparent reporting form
DOI: https://doi.org/10.7554/eLife.40655.028

### Data availability

Transcriptome sequencing and bisulfite sequencing reads are available from the European Nucleotide Archive (ENA) under accession code PRJEB26953

The following dataset was generated:

| Author(s) | Year | Dataset title | Dataset URL | Database and Identifier |
|---|---|---|---|---|
| Furci et al | 2018 | RNA-seq and WBGS | http://www.ebi.ac.uk/ena/data/view/PRJEB26953 | European Nucleotide Archive, PRJEB26953 |

The following previously published dataset was used:

| Author(s) | Year | Dataset title | Dataset URL | Database and Identifier |
|---|---|---|---|---|
| Feng S, Cokus SJ, Schubert V, Zhai J | 2014 | Genome-wide Hi-C analyses in wild type and mutants reveal high-resolution chromatin interactions in Arabidopsis | https://www.ncbi.nlm.nih.gov/sra/?term=SRP043612 | NCBI Sequence Read Archive, SRP043612 |

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
