## [Decision Letter]

Thank you for submitting your article "Identification and characterisation of epigenetic loci controlling transgenerational immune priming in Arabidopsis" for consideration by *eLife*. Your article has been reviewed by three peer reviewers, one of whom is a member of our Board of Reviewing Editors, and the evaluation has been overseen by Christian Hardtke as the Senior Editor. The reviewers have opted to remain anonymous.

The reviewers have discussed the reviews with one another and the Reviewing Editor has drafted this decision to help you prepare a revised submission.

Summary:

This work utilizes a collection of epiRILs to show that there is potential epigenetic variation in controlling defense responses.

Essential revisions:

After discussion, there were considered two essential components that need addressing to make the manuscript publishable in *eLife*.

First, it was felt that the work is highly interesting in itself but that at a number of locations, claims were made that were not supported by the data at best and occasionally contradictory information was available in other papers. As such, it was agreed that the authors should use the guidance in the reviews to rewrite the manuscript to more precisely focus on what has and has not been shown in this manuscript and others.

Second, we all agreed that the Hi-C was a potentially interesting idea but that the claims were unsupported. To support the claims would require doing Hi-C in the specific epiRILs or drop it from the manuscript at this point in time.

Reviewer #1:

– If they were just honest about what they found, it would be a really nice paper and I would support it for *eLife*. But their claims are simply unjustified by their analysis.

The authors map epiQTLs that link to resistance to *Hpa*, then assess potential epigenetic influences on transcripts and how this may be priming. Finally, the authors look for chromatin interactions in WT/*ddm1* and use them to argue that this is the mechanism. The data itself is very nice. but I find that the claims are not justified based on the conducted analysis. The experiments and data themselves will be very useful but the authors would need to focus on what they do show. To illustrate, they nicely map resistance epiQTLs and do some really nice transcriptomics. Just presenting that as plainly as what they found would be interesting.

In contrast, the writing makes extensive claims about priming, transgenerational and chromatin interactions. The transgenerational argument is trying to link in with the work showing that there is one-generation of disease effects on resistance. However, there is no analysis of if their loci really play any role in this phenomena. They just have loci whose *ddm1* marks lead to altered resistance. They would have to show that these same loci influence the transgenerational resistance phenomena to make their arguments.

The chromatin interactions are all done in *ddm1* v. WT and not in the epiRILs v. WT or v. *ddm1* so I'm not sure they can actually make their claims. Wouldn't they need to look at interactions in the epiRILs to make any of these arguments.

Priming concerns –

I should caveat the following in that I have some quant/eco/evo training in my background so I go by a very strict definition of priming; a pre-treatment leads to an elevated response to a secondary treatment that cannot be explained by any effect of the pre-treatment. Similar to the definition I work off of for epistasis. The effect of A x B is not explainable by adding up the effect of A and B separately.

Given this definition, the transcriptomics analysis does not make a direct test of priming. The authors use pairwise analysis which do not allow for a statistical testing of priming directly. To claim priming would require re-analyzing the data with a linear model where Transcript = genotype + treatment + genotype x treatment. And only the transcripts with a significant genotype x treatment term would fit into what might be called priming. Priming is the interaction term rather than the main effect of genotype or treatment.

This concern about what is or is not priming, arises from their own data that argues against priming even when they say it argues for priming. If you notice in Figure 2B, the PCA plot actually shows that the main transcript variance in the epiRILs are not priming but mainly constitutive changes as the change from Mock to *Hpa* infected is a similar level of variance in each genotype. If it was truly a predominantly priming effect, the uninduced would all be similar to the WT uninduced. Figure 2—figure supplement 2A also shows my worry about priming. If you look at the plots for what a primed transcript looks like, only 2 of those five plots would truly fit what is priming.

The authors argue that priming is the sole thing altered and that other defense traits aren't influenced. There are a couple of potential issues with this exclusionary arguments. The use of the 8 most resistance epiRIL lines to test other interactions is not really evidence of much. There isn't enough statistical power to say anything other than they aren't exactly 100% the same traits. It is possible that there is still a correlation. Additionally, at least one if not two of their loci are linked to variation in defense metabolism in this population based on previous work looking at epigenetic variation in defense metabolites within population. This shows that there are defense traits that are altered in these lines prior to infection. This previous work should be cited and incorporated appropriately when discussing the authors results as it contradicts the central claim about exclusivity of effects.

Reviewer #2:

This study uses the *ddm1* epiRIL population to identify QTL controlling resistance to. At least four epiQTL are identified which contribute >50% of the heritable variation. To explore these epiQTL further, transcriptome and methylome analyses were performed. These experiments revealed that four of the epiRIL lines with the highest resistance display a heightened transcriptome response to *Hpa* indicating these lines are already primed. This enhanced priming explains why the lines are not suffering a growth defect as there is not a constitutive response to *Hpa* or other a/biotic stresses. To understand how the priming of *Hpa* induced genes occurs the DNA methylome data were integrated, which generally revealed that the loci within the QTL regions were not directly affected by loss of DNA methylation and that perhaps there is a genome-wide trans affect occurring in these lines due to the global loss of DNA methylation. To provide a mechanistic explanation for how this could occur the authors hypothesize that long range chromatin interactions could be in play. To test this, Hi-C data from WT and *ddm1* was analyzed to revealed the existence of long range chromatin interaction between the QTL and the affected defence genes.

Overall, this is an interesting study, especially with the identification of epiRILs that are resistant to Hpa seemingly with no growth defects. However, the causal basis for these phenotypes requires additional evidence. The use of Hi-C is innovative, but it does not prove the strong statement made in the subsection “DDM1-dependent chromatin interactions between the resistance epiQTLs and distant defence genes as a potential *trans*-priming mechanism”, in the Abstract and in the Discussion.

1) Throughout the manuscript there are references to phrases that do not make sense. What are "epigenetic loci" as stated in the title or "epigenetic resistance". I would refer to these as "loci" and "resistance". Furthermore, why is "transgenerational" invoked throughout the paper? QTL are stably inherited. There are no transgenerational experiments presented in this study. Please remove this phrase from the study unless referring to the literature.

2) There is a potentially interesting observation of a trans effect being causal for the resistance, but this has not been proven as stated in the Abstract and the paper.

3) The epiQTL span regions associated with 4 out of 5 pericentromeric regions indicating that overall depletion of DNA methylation and potentially loss of heterochromatin is associated with *Hpa* resistance. What is the association between the amount of *ddm1* like chromosomes within the epiRILs and the resistance phenotype. Essentially, if you remade Figure 1 to show the amount of Col vs. *ddm1* chromosome would it correlate with the observed resistance phenotype?

4) Why was *ddm1* not included in the transcriptome study upon *Hpa* infection? This would be useful to understand if it is stronger than the epiQTL, which seems to be a prediction based on the discussion.

5) The data presented to support a "tight correlation" between hypomethylation and gene expression in Figure 3B needs additional support. While it is expected that loss of methylation and transposon genes could result in their reactivation, this is not as likely to occur at genes given that most genes are unmethylated, a small percent are gene body methylation (which is causally linked to expression) and rare genes show transposon-like methylation profiles. This can be resolved by splitting out the genes into unmethylated, gene body methylation (CG) and TE-like methylation (CG, CHG and CHH). Although some of these genes are cis-regulated by DNA methylation, it does not look like they all are as stated. This could be further tested using a scatter plot between change in DNA methylation versus change in expression.

6) There is no data provided to support the statement that "their transcriptional profile is determined by their associated TEs." Please rephrase this as a hypothesis.

7) The use of Hi-C data is an interesting idea. It is stated that "43 interactions between the epiQTLs…" were reduced or intensified in *ddm1* vs. wt. Yet, there is no statistical support to indicate whether this represents an enrichment over background expectations. This is a critical test as this forms the mechanistic basis for how the trans interactions occur.

Reviewer #3:

I think this paper is likely to be of broad interest. The demonstration that demethylation of particular bits of the genome affects a particular phenotype, in this case immunity, is intriguing. The authors have made a serious effort to understand why, and have made substantial progress in that direction. They have demonstrated increased effectiveness of callose, a known factor in resistance to this pathogen, and they have shown increased expression of genes known to be induced in response to infection. They make use of a new technology, Hi-C, and obtain data suggesting that the altered methylation may have long distance effects on expression levels of genes associated with immunity, possibly explaining the resistance phenotype. I found this rather unexpected, as conventional wisdom is that in the small genome of Arabidopsis, control of gene expression is nearly always very local. This work suggests that may not be correct, and a wider view is needed.

The authors were faced with the considerable challenge of working out how the epiQTLs affect expression of immmunity-related genes. In the last experiment, they use Hi-C and find connections between the epiQTL DNA and certain infection-inducible genes outside the QTL regions. Moreover, those genes are up-regulated in the resistant epiRILs carrying the epiQTL alleles. This is temptingly close to showing a mechanism. However, more statistical analysis is needed to make this convincing. A large number of infection-inducible genes are up-regulated in the resistant epi-RILs. Is the fraction of these detected in the Hi-C experiment as being linked to the QTL loci greater than would be expected by chance?

[Editors' note: further revisions were requested prior to acceptance, as described below.]

Thank you for resubmitting your work entitled "Identification and characterisation of hypomethylated DNA loci controlling quantitative resistance in Arabidopsis" for further consideration at *eLife*. Your revised article has been favorably evaluated by Christian Hardtke as the Senior Editor, and three reviewers.

The manuscript has been improved but there are some remaining issues that need to be addressed before acceptance, as outlined below. There are just a few requests for clarification in terminology and the like that will help to raise the readability and visibility of the manuscript.

Reviewer #1:

The authors have largely responded to my previous concerns. One thing that I think might help this version of the manuscript is to simplify the Discussion and definitions surrounding augmentation and priming. I found myself getting confused about what was being meant. It would help to have discrete explicit definitions being given to allow the reader to fully understand what the authors mean.

Reviewer #2:

The authors have made significant improvements to this study.

Reviewer #3:

This revised version of the manuscript addresses all of my concerns. I agree with the decision to leave out the Hi-C data. The inclusion of additional experiments has strengthened the work. The conclusions are now more conservative.

---

## [Author Response]

Essential revisions:After discussion, there were considered two essential components that need addressing to make the manuscript publishable in eLife.First, it was felt that the work is highly interesting in itself but that at a number of locations, claims were made that were not supported by the data at best and occasionally contradictory information was available in other papers. As such, it was agreed that the authors should use the guidance in the reviews to rewrite the manuscript to more precisely focus on what has and has not been shown in this manuscript and others.Second, we all agreed that the Hi-C was a potentially interesting idea but that the claims were unsupported. To support the claims would require doing Hi-C in the specific epiRILs or drop it from the manuscript at this point in time.

We have edited various sections in our manuscript to make sure that our claims are supported by the evidence provided. To this end, we have toned down some conclusions and/or provided additional experimental and statistical evidence to support our conclusions. In addition, we have included reference to two recent studies about the role of small RNAs in *trans*-regulation of defence gene expression (Cambiagno et al., 2018 and Liu et al.,). We also make reference to a recent study by Aller et al., 2018, who have used the same epiRIL population to study the contribution of heritable variation in DNA methylation to the production of defence-related metabolites (glucosinolates). Finally, we have removed the preliminary Hi-C analysis from the Results section of our manuscript. Our revised manuscript introduces the possibility of long-range heterochromatic interactions in the Discussion as one of multiple hypotheses by which the epiQTLs could regulate pathogen-induced expression.

Reviewer #1:If they were just honest about what they found, it would be a really nice paper and I would support it for eLife. But their claims are simply unjustified by their analysis.The authors map epiQTLs that link to resistance to Hpa, then assess potential epigenetic influences on transcripts and how this may be priming. Finally, the authors look for chromatin interactions in WT/ddm1 and use them to argue that this is the mechanism. The data itself is very nice. but I find that the claims are not justified based on the conducted analysis. The experiments and data themselves will be very useful but the authors would need to focus on what they do show. To illustrate, they nicely map resistance epiQTLs and do some really nice transcriptomics. Just presenting that as plainly as what they found would be interesting.

We thank the reviewer for her/his supportive comments. We agree that some of our assertions were too speculative or not sufficiently supported by experimental evidence. Please, see our below answers for details.

In contrast, the writing makes extensive claims about priming, transgenerational and chromatin interactions. The transgenerational argument is trying to link in with the work showing that there is one-generation of disease effects on resistance. However, there is no analysis of if their loci really play any role in this phenomena. They just have loci whose ddm1 marks lead to altered resistance. They would have to show that these same loci influence the transgenerational resistance phenomena to make their arguments.

To start with this reviewer’s concern about the term ‘transgenerational’, we used this word to refer to the hypo-methylated epiQTLs that are segregating in the epiRIL population. These loci were inherited from the *ddm1-2* parent in a wild-type background and can therefore be regarded as ‘transgenerational’. The epiRILs described in our paper were at the 8^th^ generation of self-pollination in a wild-type background (Johannes et al. 2009). In fact, Latzel et al., 2013, have shown that some hypo-methylated DNA regions from the *ddm1-2* mutant are inherited for up to 16 generations of self-pollination in a wild-type background. Since we have established a statistically significant link between the segregating resistance in the epiRIL population and four hypo-methylated epiQTLs that are inherited from the *ddm1-2* parent, we believe it is justified to refer to the resistance as ‘transgenerational’. To clarify this argument, we have adjusted the following sections in our manuscript:

“A core set of 123 epiRILs from this population at the 8th generation of self-pollination in the wild-type (Wt) background has been characterized for differentially methylated region (DMR) markers, enabling linkage mapping of heritable hypo-methylated loci controlling root growth, flowering and abiotic stress tolerance (Cortijo et al., 2014; Kooke et al., 2015).”

“In this study, we have characterised the core set of 123 lines from the *ddm1-2* x Col-0 epiRIL population for resistance against the biotrophic downy mildew pathogen *Hyaloperonospora arabidopsidis (Hpa*) to search for heritable hypo-methylated loci controlling disease resistance.”

Furthermore, we have carried out an additional experiment to confirm the transgenerational inheritance of the resistance trait and assess the ‘transgenerational’ stability of the trait. To this end. 40 F9 individuals from 8 *Hpa*-resistant lines were self-pollinated, after which the resulting F10 progenies were tested for *Hpa* resistance. As is shown in Figure 1—figure supplement 5 of the revised manuscript, the resistance was still evident in the F10 generation for every line tested, confirming that the resistance is indeed ‘transgenerational’ and passed on from the F9 to the F10 generation.

Interestingly, however, 4 epiRILs displayed statistically significant variation between the five F10 families within the line, suggesting that these epiRILs show transgenerational instability of the trait. Moreover, two of the forty F10 families showed similar levels of *Hpa* resistance as the wild-type, indicating that they had lost the transgenerational resistance completely.

“The 123 epiRILs analysed for *Hpa* resistance had been self-pollinated for 8 generations in a Wt (Col-0) genetic background since the F1 x Col-0 backcross (F9) (Johannes et al., 2009). […] Furthermore, 4 of the 8 epiRILs tested (#71, #148, #545, and #508) displayed statistically significant variation in Hpa resistance between the 5 F10 families within the epiRIL (Figure 1—figure supplement 5; Pearson’s Chi-squared test, p<0.05; † symbols), suggesting instability of the *Hpa* resistance.”

The chromatin interactions are all done in ddm1 v. WT and not in the epiRILs v. WT or v. ddm1 so I'm not sure they can actually make their claims. Wouldn't they need to look at interactions in the epiRILs to make any of these arguments.

We agree that the Hi-C analysis of the data from the Feng et al., 2014 study remains too preliminary to conclude that DDM1-dependent chromatin interactions contribute to defence priming in the resistant epiRILs. Based on the collective feedback from all reviewers and the editor, we decided to remove the Hi-C analysis from the Results section of the manuscript. The revised manuscript refers in the Discussion to the results by Feng et al., 2014, who showed that the *ddm1-2* mutation has a major effect on heterochromatic interactions in the pericentromeric regions, including the epiQTLs identified in our study. We illustrate this by referring to Figure 3—figure supplement 4, which projects the data from the Feng et al., 2014 paper onto the Arabidopsis genome and shows ample DDM1-dependent chromatin interactions originating from the epiQTLs. Considering that the same *ddm1-2* mutant was used to generate the Col-0 x *ddm1-2* epiRIL population, this provides support to the hypothesis that DDM1-dependent long-range chromatin interactions with the pericentromeric epiQTLs could play a role in *trans*-regulation of defence genes. However, we emphasize that this remains a hypothesis until further Hi-C studies with the resistant epiRILs can confirm statistically significant interactions with the constitutively primed defence genes:

“Unlike non-coding RNAs, long-range chromatin interactions can trans-regulate gene expression independently of sequence complementarity (Harmston and Lenhard, 2013; Liu et al., 2016; Weber et al., 2016; Wang et al., 2017). […] Whether these long-range interactions contribute to trans-regulation of defence gene priming would require further study, including a fully replicated Hi-C analysis of the resistant epiRILs characterised in this study.”

Priming concerns –I should caveat the following in that I have some quant/eco/evo training in my background so I go by a very strict definition of priming; a pre-treatment leads to an elevated response to a secondary treatment that cannot be explained by any effect of the pre-treatment. Similar to the definition I work off of for epistasis. The effect of A x B is not explainable by adding up the effect of A and B separately.Given this definition, the transcriptomics analysis does not make a direct test of priming. The authors use pairwise analysis which do not allow for a statistical testing of priming directly. To claim priming would require re-analyzing the data with a linear model where Transcript = genotype + treatment + genotype x treatment. And only the transcripts with a significant genotype x treatment term would fit into what might be called priming. Priming is the interaction term rather than the main effect of genotype or treatment.

Firstly, we would like to point out that there is published evidence that basal resistance in plants can be caused by an increased responsiveness of pathogen-inducible defence mechanisms, which is commonly referred to ‘constitutive priming’ (van Hulten, 2006; Ahmad et al., 2010, MPP, 11: 817-827, Ahmad et al., 2011 PCE, 34: 1191-1206, Gamir et al. 2014; Plant J, 78:227-240). This resistance phenotype has also been described for Arabidopsis mutants with reduced DNA methylation (Lopez et al., 2011; Lopez et al., 2016) and progeny from diseased parental plants that express transgenerational acquired resistance (Luna et al., 2012; Rasmann et al., 2012). Thus, the term priming is not strictly limited to describe an induced resistance response to a prior stimulus within the same generation, but can also be used to describe an innate resistance phenotype, as is the case for the epiRILs described in our current study.

Our arguments to support the conclusion that *Hpa*-resistant epiRILs are constitutively primed for defence-related gene expression is based on three statistical tests, the visualisation of the associated gene expression patterns, and the GO term analysis (which confirms enrichment of defense-related functions). The first statistical selection is based on a comprehensive linear model that includes all factors and interactions (epi-genotype, treatment, time-point and all interactions thereof). Using this model, a likelihood ratio test (LRT) selected for all differentially expressed genes by any factor and/or interaction, yielding a large number of genes. Although hierarchical cluster analysis of this gene set already indicated a large cluster displaying augmented induction in the epiRILs (Figure 2—figure supplement 1), we performed (Wald tests (+ FDR) to select for *Hpa*-inducible genes that show elevated expression in the more resistant epiRILs during *Hpa* infection (Figure 2—figure supplement 2). Wald tests are recommended by the DeSEQ2 vignette to extract specific contrasts in gene expression and allowed us e.g. to separate the *Hpa*-inducible genes from the *Hpa*-repressed genes. Using a series of Wald tests, we distinguished two *Hpa*-inducible gene groups that differ in expression profile between the Wt and epiRILs: in addition to *Hpa*-inducible genes that are constitutively up-regulated in the resistant epiRILs (Group 1), we identified a larger group of genes that show elevated levels of *Hpa*-induced expression in the resistant epiRILs (Group 2). This latter gene group was strongly enriched with defence-related GO terms at 48 hpi, which represents a critical time-point in the interaction between Arabidospis and *Hpa* (Figure 2D). Thirdly, on advice of the reviewer and the editor, we have selected all genes from the general linear model that show a statistically significant interaction between epi-genotype x *Hpa* treatment (16,009 genes), and determined the percentage of genes in Group 2 that are significant for this interaction term. As is presented in Supplementary dataset S7 in Supplementary file 1, the majority of Group 2 genes show a statistically significant interaction term for each epiRIL (> 92%), supporting a primed expression pattern. Finally, we confirm this notion by visualization of the actual expression profiles in heatmaps (Figure 2C and Figure 2—figure supplement 4), which show clearly that the vast majority of Group 2 genes exhibit augmented induction levels by *Hpa* in the resistant epiRILs.

Thus, to address this reviewer’s concerns about priming, we have applied the following changes to our manuscript:

- We have adjusted our description of the gene expression profiles. The selection for *Hpa*-inducible genes showing enhanced expression in the more resistant epiRILs now differentiates between genes that show constitutively enhanced expressed in the resistant epiRILs (now referred as “Group 1”) and genes that show elevated levels of *Hpa*-induced expression in the resistant epiRILs (now referred as “Group 2”). We have removed the notation of “primed” at this stage of the gene selection.

- On advice of the reviewer, we selected from our comprehensive general linear model all genes that show a statistically significant interaction between epi-genotype x *Hpa* treatment. This criterion alone is not sufficient to select for primed genes, since it also selects for genes that are e.g. differentially repressed by *Hpa* infection. However, when applied to the genes within Group 2, which had already been selected for enhanced levels of *Hpa*-induced expression in the epiRILs, we found that the vast majority (> 92%) of genes in this group were statistically significant for this interaction term. The combination of these criteria indicates a constitutively primed expression pattern.

- We have included z-score-normalised heatmaps of the gene expression profiles in Groups 1 and 2. This visualisation of the actual expression patterns supports our conclusion that the majority of Group 2 genes are constitutively primed in the resistant epiRILs: the genes show strongly augmented gene induction by *Hpa* in the resistant epiRILs, while showing no or only marginally higher levels of basal expression in mock-treated epiRILs (Figure 2C, Figure 2—figure supplements 3 and 4). This expression profile is consistent with the agreed definition of priming, as described by Martinez-Medina et al.,2016).

- We only refer to ‘*defence* gene priming’ after the analysis of a statistically significant interaction between epi-genotype x *Hpa* treatment, visualisation of the actual expression patterns, and the GO term analysis. The latter analysis demonstrates that Group 2 genes at the early and critical time-point of 48 hpi are strongly enriched with *defence*-related GO terms.

This concern about what is or is not priming, arises from their own data that argues against priming even when they say it argues for priming. If you notice in Figure 2B, the PCA plot actually shows that the main transcript variance in the epiRILs are not priming but mainly constitutive changes as the change from Mock to Hpa infected is a similar level of variance in each genotype. If it was truly a predominantly priming effect, the uninduced would all be similar to the WT uninduced. Figure 2—figure supplement 2A also shows my worry about priming. If you look at the plots for what a primed transcript looks like, only 2 of those five plots would truly fit what is priming.

We would like to point out that we did not conclude from the PCA plot that the epiRILs are primed for defence gene expression. What the PCA plot shows is that samples from mock- and *Hpa*-inoculated plants separate mostly along the first principal component (PC1), which explains 31% of the variation in the experiment, whereas samples from different epi-genotypes separate along the second principal component (PC2), which explains only 20% of the variation. Thus, the main transcript variation in the experiment is caused by the response of the plants to *Hpa*. Since the samples from *Hpa*-inoculated Wt plants cluster differently along PC1 than those from the *Hpa*-inoculated epiRILs, it is justifiable to conclude that the transcriptome of the epiRILs responds differently to *Hpa* infection than the Wt.

Regarding this reviewer’s concern about the expression profiles depicted in Figure 2—figure supplement 2A, we removed the label ‘Primed’ from the gene selection, which is now referred as ‘Group 2’. As detailed above, we define genes in this Group as displaying enhanced levels of *Hpa*-induced expression in the resistant epiRILs. Based on further statistical analysis of the interaction between epigenotype x *Hpa* treatment and visual confirmation of the actual gene expression profiles, we conclude that the vast majority of Group 2 genes are constitutively primed in the resistant epiRILs.

The authors argue that priming is the sole thing altered and that other defense traits aren't influenced. There are a couple of potential issues with this exclusionary arguments. The use of the 8 most resistance epiRIL lines to test other interactions is not really evidence of much. There isn't enough statistical power to say anything other than they aren't exactly 100% the same traits. It is possible that there is still a correlation.

We agree with the reviewer that the callose phenotypes and *PR1* gene expression profiles of the 8 epiRILs are insufficient to conclude that the resistance controlled by the epiQTLs is solely based on defence priming. Accordingly, we have removed the exclusionary tone about the causal link between the epiQTLs and defence priming. However, we can conclude that the 8 most resistant epiRILs in the population are constitutively primed for cell wall defence and SA-dependent *PR1* induction. This is supported by the genome-wide transcriptomic expression profiles and related GO term analysis, revealing that the majority of *Hpa*-inducible genes with defence-related functions show a constitutively primed expression pattern (see above). Furthermore, based on our population-wide screen for growth phenotypes, we can conclude that the resistance-controlling epiQTLs have no major impact on plant growth, which is consistent with the fact that defence priming has been shown to provide plants with resistance that is associated with no or minimal effects on plant growth (van Hulten et al., 2006). Thus, while we cannot exclude that other defence mechanisms have a contribution to the segregating resistance in the epiRIL population, independent lines of evidence in our study suggest that defence priming is the most plausible mechanism by which the epiQTLs control *Hpa* resistance. As is further detailed in our response to comment #7 of this reviewer (see below), we have addressed this point in the second paragraph of the Discussion section.

Additionally, at least one if not two of their loci are linked to variation in defense metabolism in this population based on previous work looking at epigenetic variation in defense metabolites within population. This shows that there are defense traits that are altered in these lines prior to infection. This previous work should be cited and incorporated appropriately when discussing the authors results as it contradicts the central claim about exclusivity of effects.

We thank the reviewer for pointing us to the Aller et al., 2018 paper; “Comparison of the Relative Potential for Epigenetic and Genetic Variation To Contribute to Trait Stability”. The revised version of our manuscript now discusses the potential contribution of pre-existing defence metabolites (glucosinolates) in the *Hpa* resistance:

“Aller et al., 2018, have recently used the same Col-0 x ddm1-2 epiRIL population to map the contribution of heritable variation in DNA methylation to the production of defence-related glucosinolate metabolites (Wibowo et al., 2016^)^. […] While we cannot exclude other mechanisms, these independent lines of evidence collectively indicate that genome-wide priming of defence genes is the most plausible mechanism by which the epiQTLs mediate quantitative disease resistance in the population.”

Reviewer #2:[…] Overall, this is an interesting study, especially with the identification of epiRILs that are resistant to Hpa seemingly with no growth defects. However, the causal basis for these phenotypes requires additional evidence. The use of Hi-C is innovative, but it does not prove the strong statement made in the subsection “DDM1-dependent chromatin interactions between the resistance epiQTLs and distant defence genes as a potential trans-priming mechanism”, in the Abstract and in the Discussion.1) Throughout the manuscript there are references to phrases that do not make sense. What are "epigenetic loci" as stated in the title or "epigenetic resistance". I would refer to these as "loci" and "resistance".

We agree that some statements and phrases in our original manuscript were unclear. While the term ‘epigenetic quantitative trait loci’ (epiQTLs) is a commonly used term to describe heritable regions of DNA hypo-methylated controlling plant complex traits, we have deleted the term ‘epigenetic’ when referring to the resistance phenotype of the epiRILs. To prevent confusion, we have also adjusted the title of manuscript: “Identification and characterisation of hypomethylated DNA loci controlling quantitative resistance in Arabidopsis”.

Furthermore, why is "transgenerational" invoked throughout the paper? QTL are stably inherited. There are no transgenerational experiments presented in this study. Please remove this phrase from the study unless referring to the literature.

We thank the reviewer for this comment. We have now included an extra experiment that addresses the transgenerational nature and stability of the *Hpa* resistance. For further details, please, see our response to comment #2 from reviewer 1.

2) There is a potentially interesting observation of a trans effect being causal for the resistance, but this has not been proven as stated in the Abstract and the paper.

We agree with the reviewer that there is no hard causal evidence to support this statement. We have therefore removed this statement from the Abstract. In addition, we have toned down the conclusiveness regarding *trans*-regulation of defence genes in our paper. Instead, we propose different mechanisms of *trans*-regulation in the Discussion as possible explanations for our finding that we did not find defence-related genes within the epiQTL regions that are *cis*-regulated by the DNA methylation (i.e. genes that show augmented expression in the epiRILs during *Hpa* infection and that are simultaneously reduced in DNA methylation).

3) The epiQTL span regions associated with 4 out of 5 pericentromeric regions indicating that overall depletion of DNA methylation and potentially loss of heterochromatin is associated with Hpa resistance. What is the association between the amount of ddm1 like chromosomes within the epiRILs and the resistance phenotype. Essentially, if you remade Figure 1 to show the amount of Col vs. ddm1 chromosome would it correlate with the observed resistance phenotype?

Each of the 123 epiRILs in the population shows different combinations of the 126 DNA methylation markers, which are stably inherited in a Mendelian fashion from the Wt and *ddm1-2* parents (Colomé-Tatché et al., 2012) Since the population is derived from a backcross, on average, the epiRIL population contains 25% of the *ddm1* haplotype and 75% of the Col-0 haplotype. Interval mapping provides the appropriate method to establish a statistical relationship between the resistance phenotype of each line and the haplotype for each marker (quantified by LOD score).

To address this reviewer’s question, we have included a supplementary dataset (S1 in Supplementary file 1), which details the DNA methylation marker distribution of the Wt and *ddm1-2* haplotypes of all epiRILs, ranked by *Hpa* resistance. The bottom row of this table shows the estimated percentage of *ddm1-2* haplotype in each epiRIL, which indeed shows a weak positive correlation with *Hpa* resistance. We attribute this trend to the size of the epiQTL intervals (particularly on chromosomes I and IV). However, as mentioned above, it is the actual interval mapping that establishes which parts of the *ddm1-2* haplotype contribute to the resistance.

4) Why was ddm1 not included in the transcriptome study upon Hpa infection? This would be useful to understand if it is stronger than the epiQTL, which seems to be a prediction based on the discussion.

We agree that including siblings of the F4 *ddm1-2* parent of the epiRIL population would have been informative. However, we did not have enough F4 seeds of this line to ensure sufficient replication and plant material for the transcriptome analysis. Because of the nature of the *ddm1-2* mutation, propagation of this mutant into a subsequent generation (F5) would increase the level of DNA hypomethylation (Kakutani et al., 1996). As such, using F5 progeny from a F4 sibling of the *ddm1-2* parent would not provide an appropriate estimation of the amount of DNA hypomethylation that had been introgressed into the wild-type background for this epiRIL population.

5) The data presented to support a "tight correlation" between hypomethylation and gene expression in Figure 3B needs additional support. While it is expected that loss of methylation and transposon genes could result in their reactivation, this is not as likely to occur at genes given that most genes are unmethylated, a small percent are gene body methylation (which is causally linked to expression) and rare genes show transposon-like methylation profiles. This can be resolved by splitting out the genes into unmethylated, gene body methylation (CG) and TE-like methylation (CG, CHG and CHH). Although some of these genes are cis-regulated by DNA methylation, it does not look like they all are as stated. This could be further tested using a scatter plot between change in DNA methylation versus change in expression.

We thank the reviewer for this helpful suggestion. The revised manuscript now includes additional scatterplots to visualise and quantify the correlation between augmented expression ratio and DNA hypomethylation of the corresponding Group 2 genes (Figure 3—figure supplement 2). For each epiQTL, linear regression analysis confirmed the positive correlations for genes in expression cluster I, which were statistically significant (Figure 3A and Figure 3—figure supplement 1). By contrast, no positive correlations could be detected for genes in expression cluster II (Figure 3A and Figure 3—figure supplement 1). This extra analysis provides statistical support to our conclusion that cluster I genes are *cis*-regulated by DNA methylation, whereas the cluster II genes are *trans*-regulated by DNA methylation.

6) There is no data provided to support the statement that "their transcriptional profile is determined by their associated TEs." Please rephrase this as a hypothesis.

We have toned done the conclusiveness of this statement and modified the sentence as follows: *“*Furthermore, analysis of the genomic context of the sixprotein-coding genes revealed the presence of overlapping and/or nearby TEs (Figure 3—figure supplement 3), *suggesting* that the correlation between augmented expression and DNA hypomethylation for these genes is determined by association with TEs.”

7) The use of Hi-C data is an interesting idea. It is stated that "43 interactions between the epiQTLs…" were reduced or intensified in ddm1 vs. wt. Yet, there is no statistical support to indicate whether this represents an enrichment over background expectations. This is a critical test as this forms the mechanistic basis for how the trans interactions occur.

We agree that additional analysis to validate a statistically significant enrichment of heterochromatic interactions between the epiQTLs and distant defence genes identified in our RNA-seq analysis would be necessary to conclude that long-range chromatin interactions contribute to priming of distant defence genes. Sadly, the publicly available dataset from Feng et al., 2014, was based on only one replicate per genotype, thereby limiting the statistical stringency of such analysis. In addition, as pointed out by the reviewer #1 and the editor, even if we would be able to provide a statistically significant justification for these specific interactions, the analysis would still only be based on the *ddm1-2* mutant, and not on the epiRILs that were analysed for genome-wide gene expression. For these reasons, we have come to the conclusion that the Hi-C analysis remains too preliminary at the current state to be presented as a main result in the manuscript.

In its current form, the Discussion of our revised manuscript raises the possibility that long-range heterochromatic interactions with peri-centromeric epiQTLs could have a contribution to *trans*-regulation of distant defence genes. We support this hypothesis by referring to the study by Feng et al., 2014, who demonstrated that the *ddm1-2* mutation has a profound effect on long-range heterochromatic interactions with pericentromeric regions, including those that are covered by our epiQTLs. To illustrate this point, we refer in the Discussion to Figure 3—figure supplement 4, which presents the interaction maps of the Wt (Col-0) and the *ddm1-2* mutant from the Feng et al., 2014 paper, and highlights the inferred DDM1-dependent interactions with our epiQTL intervals. For the reasons outlined above, we do not take this analysis further.

Reviewer #3:I think this paper is likely to be of broad interest. The demonstration that demethylation of particular bits of the genome affects a particular phenotype, in this case immunity, is intriguing. The authors have made a serious effort to understand why, and have made substantial progress in that direction. They have demonstrated increased effectiveness of callose, a known factor in resistance to this pathogen, and they have shown increased expression of genes known to be induced in response to infection. They make use of a new technology, Hi-C, and obtain data suggesting that the altered methylation may have long distance effects on expression levels of genes associated with immunity, possibly explaining the resistance phenotype. I found this rather unexpected, as conventional wisdom is that in the small genome of Arabidopsis, control of gene expression is nearly always very local. This work suggests that may not be correct, and a wider view is needed.The authors were faced with the considerable challenge of working out how the epiQTLs affect expression of immmunity-related genes. In the last experiment, they use Hi-C and find connections between the epiQTL DNA and certain infection-inducible genes outside the QTL regions. Moreover, those genes are up-regulated in the resistant epiRILs carrying the epiQTL alleles. This is temptingly close to showing a mechanism. However, more statistical analysis is needed to make this convincing. A large number of infection-inducible genes are up-regulated in the resistant epi-RILs. Is the fraction of these detected in the Hi-C experiment as being linked to the QTL loci greater than would be expected by chance?

We agree with the reviewer that our initial Hi-C analysis remains too preliminary to conclude a contribution of DDM1-dependent heterochromatic interactions to defence gene regulation in the resistant epiRILs. As outlined in our response to comment #3 from the first reviewer and comment #7 from the second reviewer, additional statistical analysis would be needed to support this claim. Sadly, the publicly available dataset from Col-0 and *ddm1-2* plants is based on one replicate per genotype, so lacks statistical power to make this analysis more conclusive. Moreover, the analysis would still only be based on the *ddm1-2* mutant, and not on the epiRILs that were analysed for genome-wide gene expression. Ultimately, we believe that a fully replicated Hi-C analysis on the *Hpa*-resistant epiRILs compared to Wt would be needed to provide more conclusive statistical evidence for long-range chromatin interactions with the genes identified in from RNA-seq analysis.

For these reasons, we have omitted the Hi-C analysis from the main Results section of the manuscript. However, the Discussion of our revised manuscript addresses the possibility that long-range chromatin interactions could play a role in the *trans*-regulation of global defence gene expression by the epiQTLs. We support this hypothesis by referring to the results in the Feng et al., 2014 study, which demonstrate that the *ddm1-2* mutation has a profound impact on the long-range heterochromatic interactions with pericentromeric regions, including those that are covered by our epiRIL regions. For details, see our response to comment #3 of reviewer 1.

[Editors' note: further revisions were requested prior to acceptance, as described below.]

Reviewer #1:The authors have largely responded to my previous concerns. One thing that I think might help this version of the manuscript is to simplify the Discussion and definitions surrounding augmentation and priming. I found myself getting confused about what was being meant. It would help to have discrete explicit definitions being given to allow the reader to fully understand what the authors mean.

The Introduction of our revised manuscript provides a better definition of priming, and how priming relates to augmentation of defence-related gene induction and quantitative resistance:

“Recent evidence has suggested that reduced DNA methylation increases the responsiveness of the plant immune system (Espinas, Saze and Saijo, 2016. […] In some cases, priming of defense-related genes is associated with post-translational histone modifications that mark a more open chromatin structure (Jaskiewicz, Conrath and Peterhänsel, 2011; Luna et al., 2012).”